# Cortical waves mediate the cellular response to electric fields

**Qixin Yang[1,2], Yuchuan Miao[3], Leonard J Campanello[1,2], Matt J Hourwitz[4], Bedri Abubaker-Sharif[3], Abby L Bull[1,2], Peter N Devreotes[3], John T Fourkas[2,4], Wolfgang Losert[1,2]\***

[1]Department of Physics, University of Maryland, College Park, United States; [2]Institute for Physical Science and Technology, University of Maryland, College Park, United States; [3]Department of Cell Biology, Johns Hopkins University, Baltimore, United States; [4]Department of Chemistry & Biochemistry, University of Maryland, College Park, United States

**\*For correspondence:** wlosert@umd.edu

**Competing interest:** The authors declare that no competing interests exist.

**Abstract** Electrotaxis, the directional migration of cells in a constant electric field, is important in regeneration, development, and wound healing. Electrotaxis has a slower response and a smaller dynamic range than guidance by other cues, suggesting that the mechanism of electrotaxis shares both similarities and differences with chemical-gradient-sensing pathways. We examine a mechanism centered on the excitable system consisting of cortical waves of biochemical signals coupled to cytoskeletal reorganization, which has been implicated in random cell motility. We use electro-fused giant *Dictyostelium discoideum* cells to decouple waves from cell motion and employ nanotopographic surfaces to limit wave dimensions and lifetimes. We demonstrate that wave propagation in these cells is guided by electric fields. The wave area and lifetime gradually increase in the first 10 min after an electric field is turned on, leading to more abundant and wider protrusions in the cell region nearest the cathode. The wave directions display 'U-turn' behavior upon field reversal, and this switch occurs more quickly on nanotopography. Our results suggest that electric fields guide cells by controlling waves of signal transduction and cytoskeletal activity, which underlie cellular protrusions. Whereas surface receptor occupancy triggers both rapid activation and slower polarization of signaling pathways, electric fields appear to act primarily on polarization, explaining why cells respond to electric fields more slowly than to other guidance cues.

## Editor's evaluation

The authors combine a series of clever biological approaches to fuse small *Dictyostelium* cells into "giant cells" that greatly facilitate the spatial resolution of actin wave dynamics without or with electrical stimulation when grown on smooth or nano-textured surfaces. This compelling experimental system opens possibilities for the field to analyze the molecular subtleties involved in these cytoskeletal reorganizations.

## Introduction

Electrotaxis, which refers to the directed migration of cells under the guidance of an electric field (EF), is important in wound healing, development, and regeneration (*Cortese et al., 2014*; *Lin et al., 2008*; *Zhao et al., 2006*). EFs have been shown to cause several key signaling molecules to be distributed asymmetrically across cells (*Sato et al., 2009*; *Zhao et al., 2002*; *Zhao et al., 2006*), setting up cell polarity. The one-order-of-magnitude range of EF strengths sensed by cells (*Zhao et al., 2002*) is considerably smaller than the four-orders-of-magnitude concentration sensitivity in chemotaxis

(*Harvath, 1991*). Furthermore, whereas cells respond to chemical guidance cues on a time scale of seconds and develop polarity over several minutes, the response to an EF can take up to 10 min or more after the EF is turned on *Wang et al., 2014*; *Zhao et al., 2006*. These differences raise the possibility that the rapid gradient sensing mechanisms do not serve as primary mediators of EF sensing by cells. In this study, we examine whether, after turning on an EF, the gradual polarization of the excitable biochemical networks that organize actin polymerization comprises a slow-acting mediator of the cellular response to the EF.

Actin polymerization, coordinated with its associated signaling molecules, self-organizes into microscale spatial regions that travel as waves across plasma membranes. These waves drive various cell behaviors, such as migration and division (*Bhattacharya et al., 2019*; *Bretschneider et al., 2009*; *Flemming et al., 2020*; *Gerhardt et al., 2014*; *Gerisch, 2010*). The wave system can be described as a coupled signal-transduction excitable network – cytoskeletal excitable network (STEN-CEN) (*Devreotes et al., 2017*; *Miao et al., 2019*). STEN-CEN has the characteristics of an excitable system, including exhibiting an activation threshold for wave initiation and experiencing refractory periods. It has been shown that the STEN-CEN wave properties dictate protrusion properties (*Miao et al., 2019*). Tuning the activity levels of key components in STEN-CEN changes wave patterns, which leads to the transition of protrusion profiles. An activator/inhibitor, reaction/diffusion system model successfully recapitulates the experimental results (*Bhattacharya et al., 2020*; *Bhattacharya and Iglesias, 2019*). For simplicity, here we will refer to STEN-CEN waves as cortical waves.

One challenge in investigating whether cortical waves can act as the mediators of EFs is that in many of the cell types that show a strong response to EFs, the wave area is comparable to the cell area. Furthermore, waves are generated at the leading edge of the cell during directed migration (*Xu et al., 2003*), so that wave dynamics are tightly coupled with cell dynamics. For instance, when a cell responds to an EF reversal, waves typically remain at the cell front as the cell turns. It is not known whether the waves drive cells to turn or the cell polarity keeps the previous leading edge more active so that this edge responds first.

To distinguish between wave response and cell motion, we use electro-fused giant *D. discoideum* (*Neumann et al., 1980*) with diameters up to ten times larger than that of an individual cell. Multiple simultaneous waves can be generated across the surface contact area of a giant cell (*Gerhardt et al., 2014*). These waves also generate actin-filled macropinosomes on the dorsal membrane (*Veltman et al., 2016*). The giant cells provide an excellent opportunity to study cortical wave dynamics in multiple cell regions simultaneously.

We further use nanotopography to alter the waves' spatial structures and characteristic timescales. Upon contact with nanotopography, cells produce quasi-1D wave patches. The phenomenon of guided actin polymerization by nanotopography is known as esotaxis (*Driscoll et al., 2014*), and has been investigated in detail (*Ketchum et al., 2018*; *Lee et al., 2020*). There are several advantages of incorporating nanotopography in our study. First, these waves persist for a shorter time on nanotopography than on flat surfaces, enabling us to investigate whether wave systems with different characteristic timescales respond to EFs differently. Second, due to the shorter lifetime, waves on ridged surfaces only propagate in local regions of giant cells. Therefore, nanotopography allows us to distinguish between local and global mediation of the EF response.

## Results

### Cortical waves and cell migration can be studied independently in giant cells

We imaged cells that simultaneously expressed both limE-RFP and PH$_{Crac}$-GFP. The former allows us to monitor filamentous actin (F-actin), which represents CEN activities. The latter enables us to monitor phosphatidylinositol-3,4,5-trisphosphate (PIP3), an indicator of STEN activities. In single, differentiated cells, usually only one wave is generated at the leading edge (*Figure 1a* and *Video 1*), and the wave motion is coupled with cell motion. For instance, when the cell in *Figure 1a* changed its direction of motion, the wave remained at the leading edge (72 s - 120 s).

In giant cells, multiple waves were initiated randomly and propagated radially across the basal membranes (*Figure 1b* and *Video 2*). CEN is driven by STEN, but has a substantially shorter characteristic timescale. Thus, PIP3 waves displayed band-like shapes, whereas F-actin appeared across the

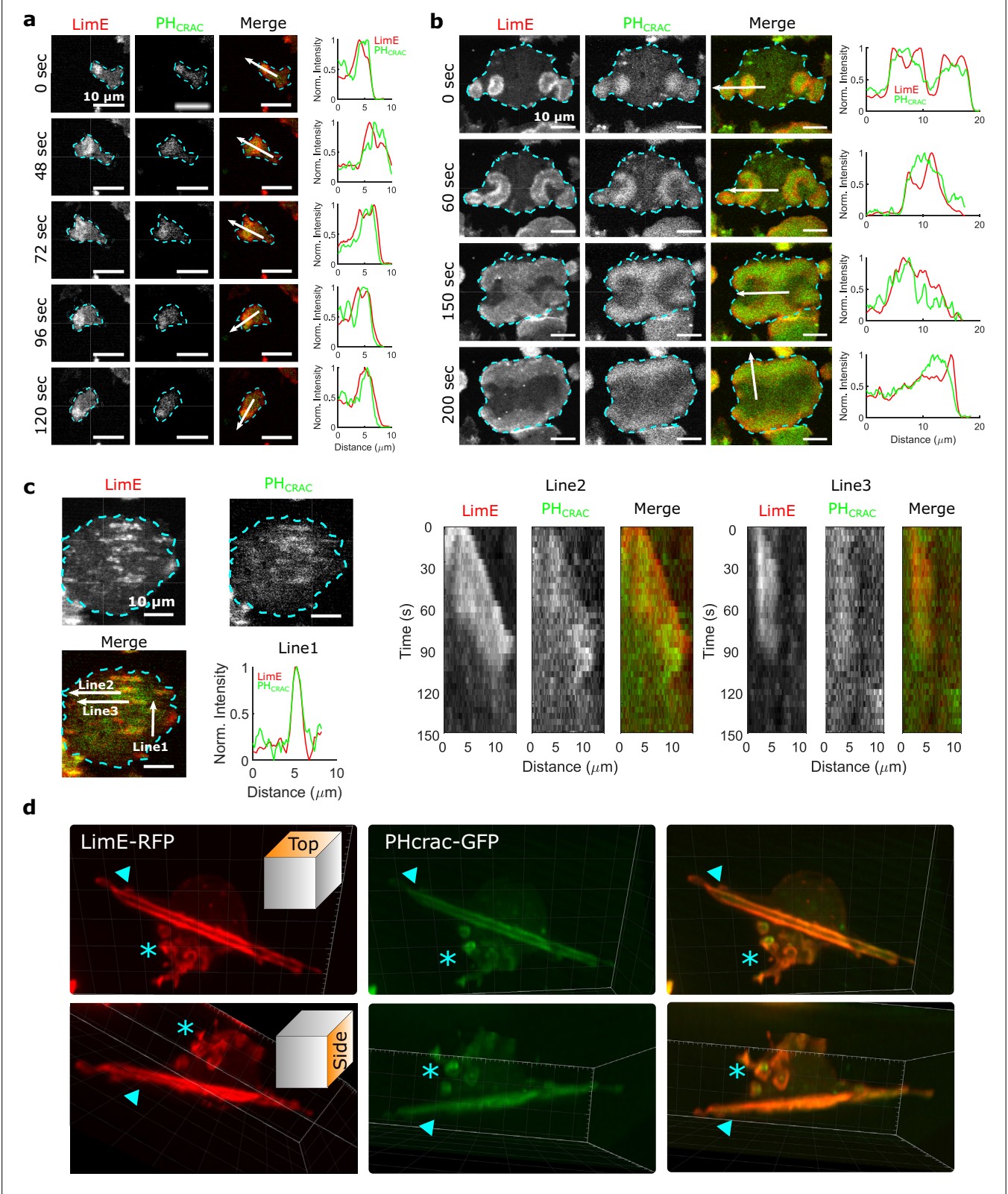

**Figure 1.** STEN-CEN waves in single cells and giant cells.

(a) Snapshots of a differentiated, single *D. discoideum* cell expressing limE-RFP and PHcrac-GFP, with cell boundaries denoted with blue dashed lines. The right column shows the normalized intensity of limE and PHcrac from the arrows in the merge images. The scale bars are 10 μm. (b). Snapshots of an electrofused giant *D. discoideum* cell on a flat surface, with scanning profiles in the right column. All scale bars are 10 μm. (c) A snapshot of an

*Figure 1 continued on next page*

Figure 1 continued

electrofused giant cell on the ridged surface. The left kymographs are from the line 2 and line 3 specified in the merged image. Line two shows a wave propagating along nanoridges, and line three shows a wave that existed briefly and then dissipated. (**d**) 3D reconstruction (single time point) of a single *D. discoideum* cell plated on nanoridges, acquired using a lattice light-sheet microscope. Here, we show the top aspect view (top row) and the side aspect view (bottom row). On the dorsal membrane of the cell, there are waves forming microcytotic cups (triangle) on the curved membrane, and on the basal membrane, there are streak-like waves (asterisk). The red channel represents limE-RFP, and the green channel represents PHcrac-GFP. As both the side and top views show, the dorsal waves and basal waves are independent structures, but both are composed of coordinated F-actin and PIP3.

The online version of this article includes the following figure supplement(s) for figure 1:

**Figure supplement 1.** Colocalization of PIP3 and F-actin in an EF.

bands with higher levels at the rims of PIP3 waves (*Miao et al., 2019*). As shown in *Figure 1b*, colliding waves did not cross, but instead rotated by 90° (*Figure 1b*, 150 s - 200 s). This behavior is suggestive of a refractory period following excitation, which is a hallmark of an excitable system. On nanoridges, the giant cells generated multiple, quasi-1D patches of F-actin and PIP3 with shorter lifetimes than on flat surfaces (*Figure 1c* and *Video 3*). Some waves formed and propagated for a short distance (Line two in *Figure 1c*), whereas others formed and then quickly dissipated (Line three in *Figure 1c*). The wave dissipation can be explained in terms of an excitable system with lateral inhibition, in which the dispersion of the inhibitor is faster than that of the activator. Thus, the waves eventually dissipate due to the spatial accumulation of the inhibitor. Prior studies have shown that in this situation, the excitable system threshold determines the wave duration (*Bhattacharya et al., 2020*; *Ermentrout et al., 1984*). As was the case on flat surfaces, 1D patches occurred throughout the basal surfaces on ridges, and thus were independent of cell motion.

Waves were also generated on the dorsal planes. In contrast to basal waves, which propagated across the surface contact (*Videos 2 and 3*), dorsal waves were associated with membrane deformations, and resembled macropinosomes (*Video 4*). Based on 3D lattice light-sheet images of a cell plated on nanoridges (*Figure 1d* and *Video 5*), activation of PIP3 and F-actin was correlated in both basal waves and dorsal waves. However, the dorsal waves were primarily generated in cuplike structures, whereas the stripe-like basal waves spanned the entire basal plane. In all cases, PIP3 activity was coordinated with F-actin activity (Profiles in *Figure 1a, b and c*), both in the absence (*Figure 1*) and presence of an EF (*Figure 1—figure supplement 1*, *Video 6*). Therefore, in the experiments described below, we only monitored F-actin activity.

## EFs increase the area, duration, and speed of waves on nanoridges

We found that giant cells respond to a narrow range (15 V/cm to 20 V/cm) of EF amplitudes (*Videos 7 and 8*), and that higher voltage (35 V/

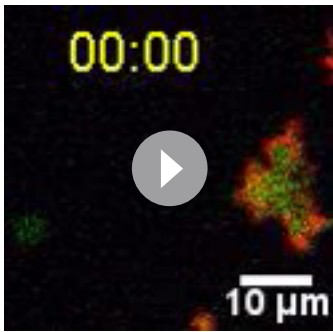

**Video 1.** Time-lapse confocal videos of PH$_{Crac}$ (green) and limE (red) on the basal surface of a single, differentiated *Dictyostelium discoideum* cell set on a flat surface. Images were acquired every 4 s and shown at 5 frames/s.

https://elifesciences.org/articles/73198/figures#video1

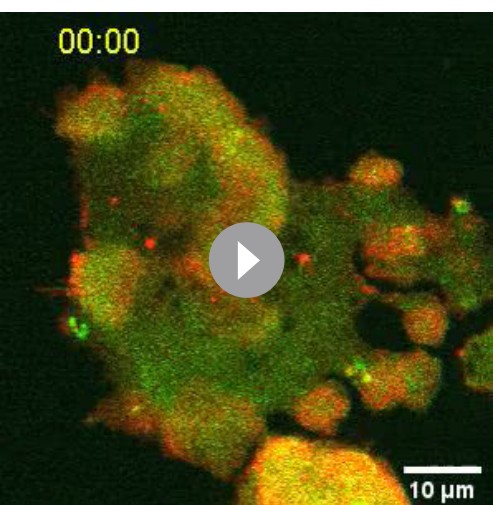

**Video 2.** Time-lapse confocal videos of PH$_{Crac}$ (green) and limE (red) on the basal surface of a giant *Dictyostelium discoideum* cell set on a flat surface. Images were acquired every 10 s and shown at 5 frames/s.

https://elifesciences.org/articles/73198/figures#video2

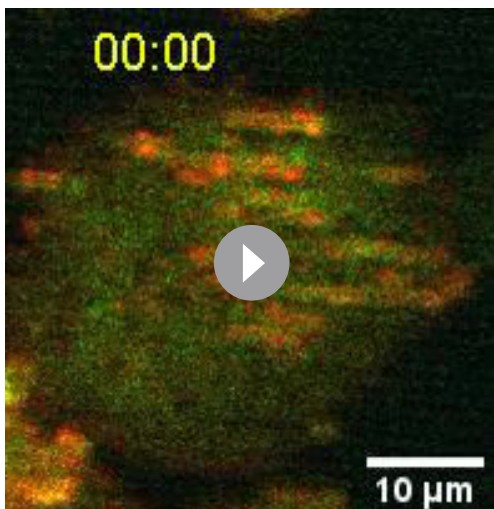

**Video 3.** Time-lapse confocal videos of PH$_{Crac}$ (green) and limE (red) on the basal surface of a giant *Dictyostelium discoideum* cell set on a ridged surface. Images were acquired every 10 s and shown at 5 frames/s.
https://elifesciences.org/articles/73198/figures#video3

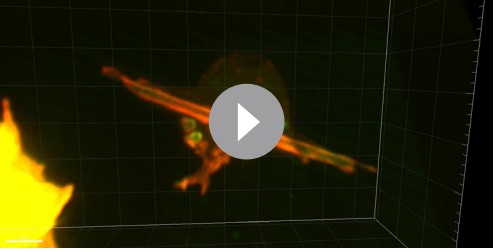

**Video 5.** 3D reconstruction of Lattice LightSheet data of PH$_{Crac}$ (green) and limE (red) of a single, vegetative *Dictyostelium* discoideum cell set on a ridged surface.
https://elifesciences.org/articles/73198/figures#video5

cm) damaged cells. The 1D waves generated on nanoridges related to esotaxis enabled us to quantify the effects of a 20 V/cm EF (all EFs used here are of this magnitude, see *Figure 2—figure supplement 1* and Materials and methods for more details about the electrotaxis experiments) on the areas, durations, and speeds of waves. *Figure 2a* shows snapshots of the dynamics of F-actin in a giant cell on parallel nanoridges with a 1.6 µm spacing. In the absence of an EF (top row in *Figure 2a*), individual actin polymerization events were initiated in patches on the basal surfaces (*Figure 2a* and *Video 9*).

In the first several minutes after turning on the EF, most patches propagated as a wave along a single ridge (*Figure 2a*, blue inset). After the EF was on for 10 min, some patches appeared to undergo coordinated motion across several ridges (*Figure 2a*, pink inset). We calculated the ratio of F-actin occupancy to cell area, and found that an EF increased the overall level of actin polymerization by a factor of two to three (*Figure 2b*). Actin patches were larger in the presence of an EF and organized into larger groups located preferentially at the cell front (bottom row in *Figure 2a*, 20 min, and 25 min), leading to wider protrusions at cell fronts that drove directed cell migration (*Figure 2—figure supplement 2*). To determine whether the groups comprised a single, large wave growing across multiple ridges or multiple, small patches nucleated in close proximity, we measured the dynamics of the patches using optical flow (*Lee et al., 2020*), focusing on the patch edges (*Figure 2c*, left image. See Materials and methods for more details). If both edges of a patch were moving in the same direction, the

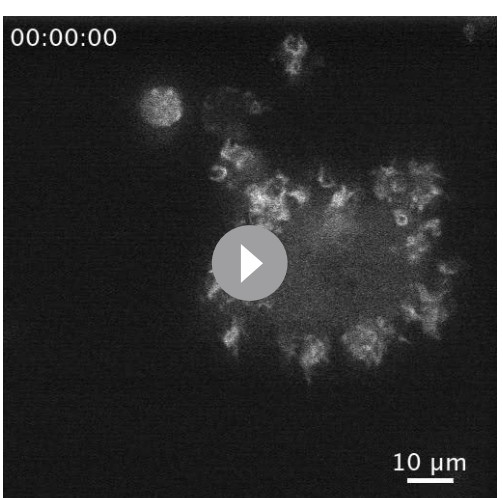

**Video 4.** Time-lapse confocal videos of limE-RFP on the dorsal membrane of a giant *Dictyostelium discoideum* cell. Images were acquired every 10 s and shown at 5 frames/s.
https://elifesciences.org/articles/73198/figures#video4

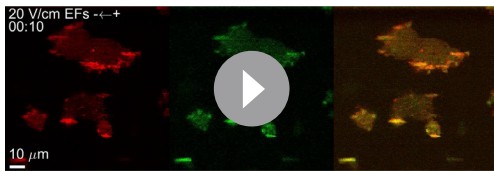

**Video 6.** Time-lapse confocal videos of PH$_{Crac}$ (green) and limE (red) on the basal surface of a giant *Dictyostelium* discoideum cell set on a ridged surface. An electric field of 20 V/cm was applied at 0 min, where the cathode was set at the left side. Then the field was reversed to cathode being on the right at 35 min. Images were acquired every 10 s and shown at 5 frames/s.
https://elifesciences.org/articles/73198/figures#video6

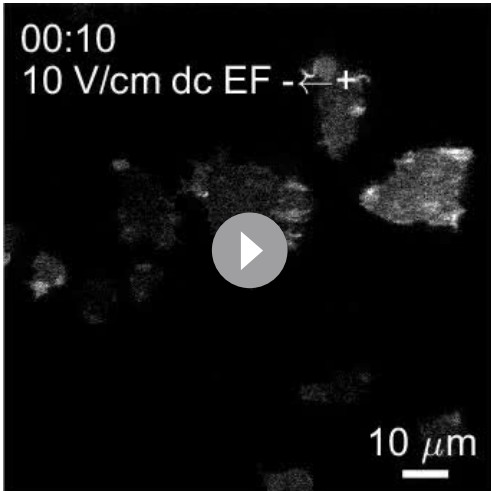

**Video 7.** Time-lapse confocal videos of LimE-RFP on the basal surface of a giant *Dictyostelium* discoideum cell set on a ridged surface. An electric field of 10 V/cm was applied at 0 min, where the cathode was set at the left side. Images were acquired every 10 s and shown at 5 frames/s.

https://elifesciences.org/articles/73198/figures#video7

structure was classified as a single, large wave. If the edges were not coordinated, the patch was classified as multiple, individual structures. This method enabled us to capture accurately waves that span across multiple ridges and are moving coordinately.

Once the large actin structures were classified, their instantaneous dimensions were measured parallel and perpendicular to the ridges (*Figure 2c*). Density scatter plots of both dimensions exhibit elliptical contours (*Figure 2d*), suggesting that nanotopography constrains wave growth. With an EF parallel to the ridges, the waves broadened in both directions (*Figure 2d*). The average increases in wave dimension parallel and perpendicular to the ridges were 20% and 13%, respectively, and the average increase in wave area was 44%. An increase in wave duration was also observed, with the minimum wave area correlated to the duration (*Figure 2e*, black circles). The wave area depends exponentially on the maximum wave duration (*Figure 2e*, solid black lines), allowing us to extract a characteristic wave time scale via

$$Area_{min} = C * e^{\frac{Duration}{T}} \qquad . \qquad (1)$$

Here, $C$ is a constant, and $T$ is the characteristic time scale, which is 48 s with no EF and 61 s in the presence of a 20 V/cm EF. This difference is consistent with the EF drawing the system closer to the excitability threshold. An average increase of 9% in wave propagation speed was also observed in the presence of an EF (*Figure 2f*).

## EFs guide the direction of actin waves

Next, we consider the directional guidance of actin waves by EFs on nanoridges (*Figure 3a* and *Video 9*) and on flat surfaces (*Figure 3b* and *Video 10*, see Methods for details). The EF was introduced at 0 min ($T_1$), and in the first 2 min had little effect on the actin dynamics on any surface. On nanoridges, actin waves continued to propagate preferentially along the ridges (*Figure 3a*, $T_1$). On flat surfaces, the waves propagated radially in groups, as seen from the broad distribution at $T_1$ in *Figure 3b*. In the presence of an EF, the waves propagated preferentially towards the cathode within ~15 min (*Figure 3a and b*, $T_2$). The perpendicular spread was significantly more limited on nanoridges (*Figure 3a*, $T_2$).

The direction of the EF was reversed after the cell had commenced steady directional migration, which took ~20–25 min. Following the field reversal, waves on ridged surfaces reoriented toward the new cathode within 5 min (*Figure 3a*, $T_3$). On flat surfaces, the wave propagation direction was perpendicular to both the previous and

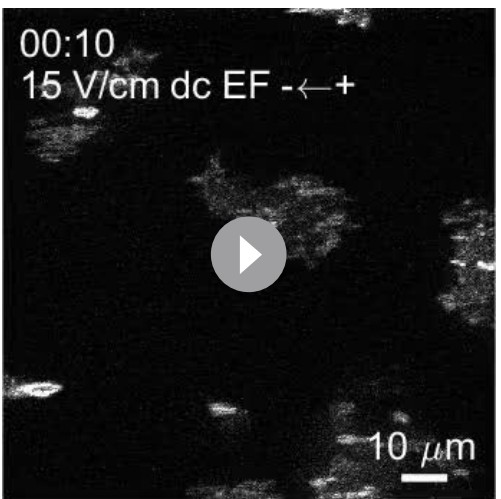

**Video 8.** Time-lapse confocal videos of LimE-RFP on the basal surface of a giant *Dictyostelium* discoideum cell set on a ridged surface. An electric field of 15 V/cm was applied at 0 min, where the cathode was set at the left side. Images were acquired every 10 s and shown at 5 frames/s.

https://elifesciences.org/articles/73198/figures#video8

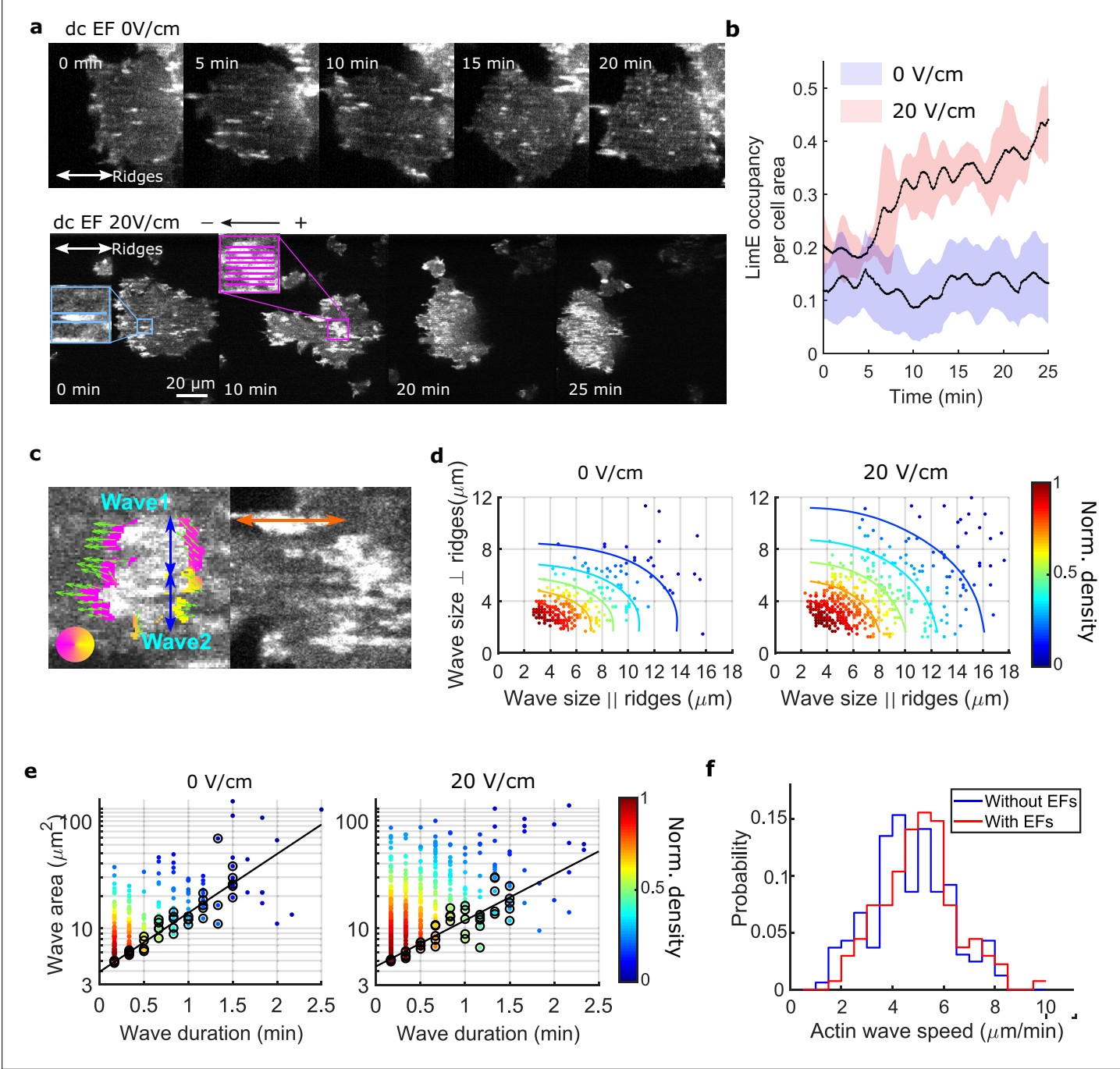

**Figure 2.** EFs alter F-actin wave properties.

(a) limE images of a giant cell on nanoridges without an EF (top) and in a 20 V/cm EF turned on at 0 min (bottom). (b). The temporal change of the percentage of the cell area occupied by limE without an EF (blue, $N_{cell}$ = 5) and in a 20 V/cm EF introduced at 0 min (red, $N_{cell}$ = 4). The shaded areas represent the mean plus or minus one standard deviation. (c). Division of groups of waves. The color represents the orientation of optical-flow vectors according to the color wheel. The green arrows are the optical-flow vectors, the length of which correspond to the magnitude of motion. The left image is an example of a large structure composed of two independent substructures, where the vectors at the right edge are not moving in the same direction. The wave scales in the directions perpendicular to (blue arrows) and parallel to the ridges (orange arrow) were measured on the preprocessed waves. (d) Density scatter plots of wave scales parallel to ridges vs. perpendicular to ridges. (e) Density scatter plots of actin-wave dimension vs. actin-wave duration. For each wave duration, the five points with the smallest wave areas (black circles) were selected to fit the boundaries (solid black lines). (f) Distributions of wave propagation speeds before (blue, $N_{wave}$ = 125) and after (red, $N_{wave}$ = 163) applying an EF. The analyses in d-f were based on N = 4 independent experiments. The two distributions are different (Two-sample t-test, p = 0.017).

The online version of this article includes the following source data and figure supplement(s) for figure 2:

*Figure 2 continued on next page*

*Figure 2 continued*

**Source data 1.** LimE occupancy normalized by cell area in 0 V/cm and 20 V/cm EF.

**Source data 2.** Wave size parallel/ perpendicular to ridges.

**Source data 3.** Wave duration and wave area.

**Source data 4.** Wave propagation speed.

**Figure supplement 1.** Schematic of the 3D-printed chamber used for electrotaxis experiments.

**Figure supplement 2.** EFs induce keratocyte-mode migration, producing larger protrusions at cell fronts.

the new EF directions at ~7 min after the field reversal (*Figure 3b*, T₃). Preferential propagation toward the new cathode occurred after ~13 min (*Figure 3b*, T₄). The difference in response time between nanoridges and flat surfaces may be related to the fact that waves persist longer on flat surfaces than on nanoridges (*Figure 3c*).

*Figure 3d* shows the continuous temporal changes of the orientation distributions. On nanoridges, the preferred wave directions switched directly following EF reversal (left plot in *Figure 3d*), whereas on flat surfaces waves maintained a preferred direction that changed continuously in a U-turn behavior (Right plot in *Figure 3d*). In contrast, although single *D. discoideum* cells undergo U-turns in response to EF reversal (*Sato et al., 2007*), giant cells did not (*Video 10*).

Wave turning may be related to differences in the patterns of wave expansion (*Figure 3e*). On flat surfaces, waves started from a small patch (*Figure 3e*, S1) and eventually broke into band-shaped waves (*Figure 3e*, S4). During directed migration, the intermediate expansion of actin waves (*Figure 3e*, S2, S3 in the top row) was biased by the EF, resulting in band-shaped waves propagating preferentially towards the cathode (*Figure 3e*, S4 in top row). After EF reversal, waves expanded in all directions (*Figure 3e*, S2, S3 in bottom row), such that optical-flow analysis captured turning behavior more frequently.

We also simultaneously imaged limE-RFP at the basal plane (near the surface contact) and the dorsal plane (6 μm higher). Dorsal waves (*Videos 11 and 12*) are localized at cell fronts and rear-ranged to the new fronts following EF reversal (*Figure 3—figure supplement 1*). Rather than directly switching preferential direction, dorsal waves gradually turned toward the new cathode (*Figure 3—figure supplement 1*), in a manner similar to that of basal waves on flat surfaces. Thus, two different response times to EF reversal exist within the same cell, with a faster response for basal waves guided by nanotopography and a slower response for the free dorsal waves. On flat surfaces, the two responses are synchronized (*Figure 3—figure supplement 2*).

## Subcellular spatial inhomogeneity of the response to EFs on nanoridges

Although waves in migrating *D. discoideum* cells localize predominantly at the leading edge (*Weiner et al., 2007*; *Zhao et al., 2002*), waves are observed across the basal layer in giant *D. discoideum* cells. We analyzed the smaller, shorter lived waves on nanoridges. Although the wave locations were distributed essentially uniformly throughout cells in the absence of an EF, more waves were generated at the cell fronts in the presence of an EF (*Figure 4a* and *Video 9*). In addition, the average area per wave was larger near the front of cells in an EF (*Figure 4b*). We also measured the wave properties in the single

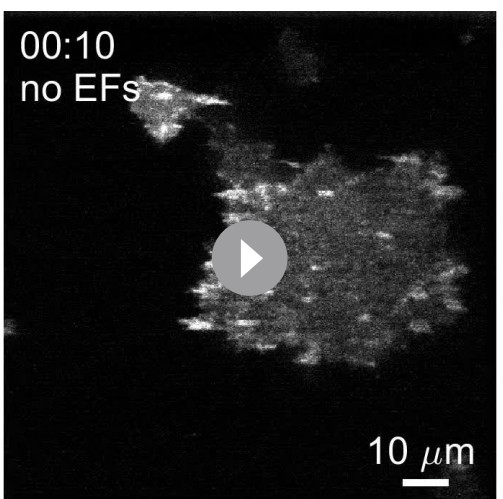

**Video 9.** Time-lapse confocal videos of LimE-RFP on the basal surface of a giant *Dictyostelium* discoideum cell set on a ridged surface. An electric field of 20 V/cm was applied at 0 min, where the cathode was set at the left side. Then the field was reversed to cathode being on the right at 30 min. Images were acquired every 10 s and shown at 5 frames/s.
https://elifesciences.org/articles/73198/figures#video9

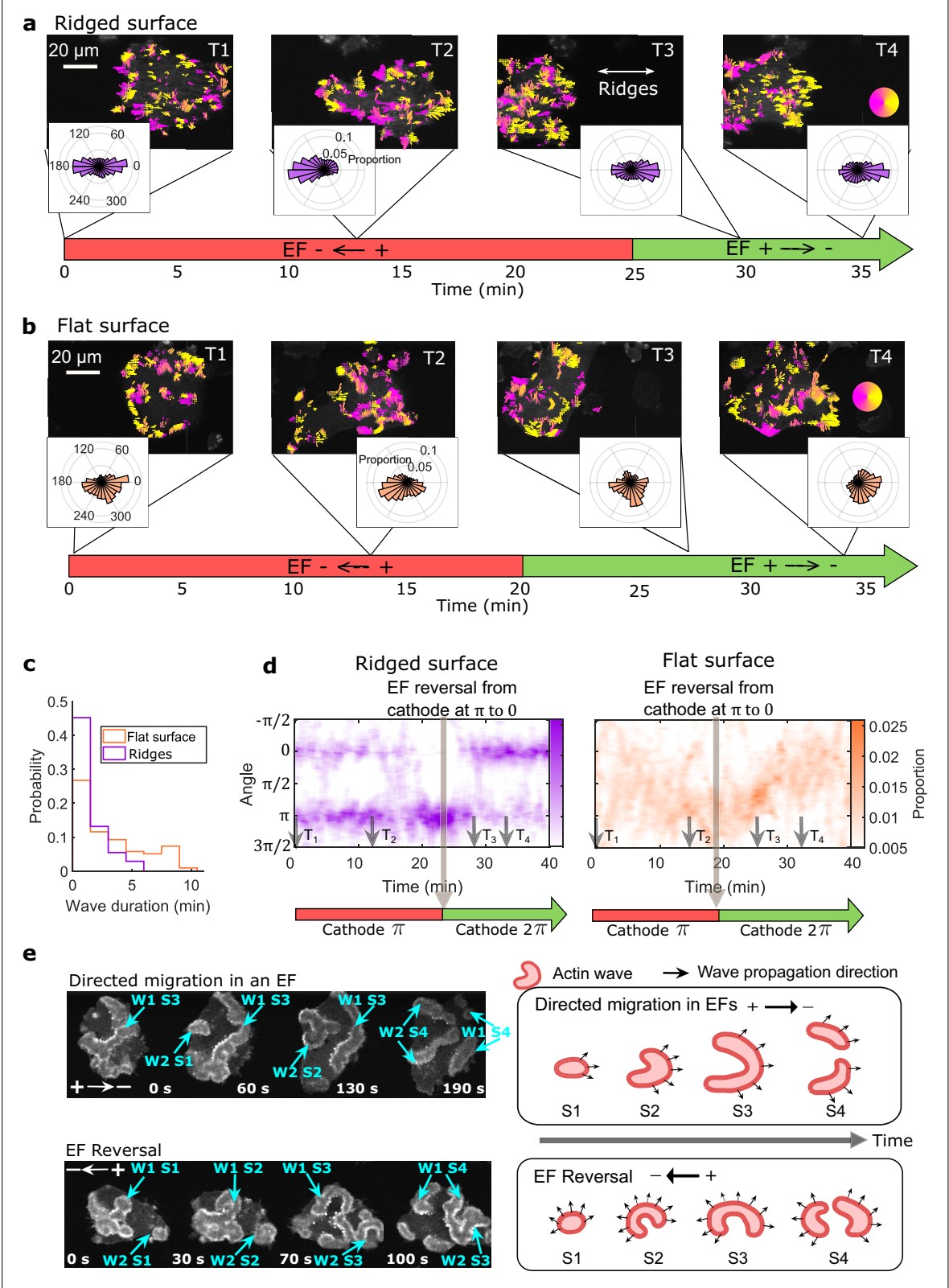

**Figure 3.** EFs guide actin waves. (**a, b**) Optical-flow analysis of actin-wave dynamics in giant cells on ridged and flat surfaces. The top row shows a time series of limE images for giant cells overlaid with optical-flow vectors, the color of which is coded according to the color wheel. The accompanying polar plots show the corresponding orientation displacements of optical-flow vectors. For both a and b the EF was turned on at 0 min. The bottom time stamp indicates when the EF was reversed from the cathode being on the right (red) to the cathode being on the left (green). (**c**) Distributions of wave

*Figure 3 continued on next page*

*Figure 3 continued*

duration from three independent days of experiments. The distributions were weighted by wave area, because the number of long-lasting large waves on flat surfaces ($N_{wave}$ = 359) is smaller than the number of short-lived small patches on ridged surfaces ($N_{wave}$ = 658). Correspondingly, the absolute waves counts do not match the pixel-based, optical-flow analysis in a and b. Based on a two-sample t-test on the wave areas on flat surfaces vs. on ridges, the null hypothesis was rejected at the 5% significance level with p = 2 × 10$^{-15}$. (**d**). Kymographs of orientation displacements of optical-flow vectors. The x-axes of the kymographs represent time, and the y-axes represent orientation. The colors represent the proportions. The EF was turned on at time $T_1$, and was reversed at the time denoted by the black arrow (**e**) LimE snapshots showing the patterns of actin-wave expansion during steady directed migration in a constant EF (top) and after reversing the EF direction (bottom). The blue arrows point to specific stages of wave expansion. W: Wave, S: Stage of wave expansion. The right panel is a cartoon illustrating the patterns of actin-wave expansion during directed migration in EFs (top) and after EFs were reversed (bottom).

The online version of this article includes the following source data and figure supplement(s) for figure 3:

**Source data 1.** Wave duration on flat surfaces / ridged surfaces.

**Figure supplement 1.** Basal actin waves reverse direction on nanoridges, whereas dorsal waves turn.

**Figure supplement 2.** Basal and dorsal waves on flat surfaces both turn in response to EF reversal.

cells scattered throughout the field of view but did not observe a corresponding gradient of wave properties among single cells closer to the cathode versus the cells closer to the anode. This result indicates that the spatial inhomogeneity shown in *Figure 4a and b* was caused by the EF rather than by the absolute electrical potential relative to the ground (*Figure 4—figure supplement 1*).

We explored the response of this inhomogeneity to EF reversal by tracking each wave location relative to the cell centroid in the 12 min following EF reversal (*Figure 4c*). New waves started to appear near the side of the cell facing the new cathode within 3 min (*Figure 4c*, left region of $P_2$), whereas the complete inhibition of wave generation near the old cell front took longer (*Figure 4c*, right region of $P_5$). This observation suggests that the initiation at a new cell front and the inhibition of waves at the old front are regulated by two distinct processes with different timescales.

Next, we looked at the time required to switch propagation direction in different subcellular regions following EF reversal. The basal membrane was segmented into an 'old front' region (facing the original cathode) and a 'new front' region (facing the new cathode), as illustrated in *Figure 4d*. The distributions of wave propagation directions show that waves in the new front region switched their preferential direction at ~4 min. In contrast, waves in the old front region changed their preferential

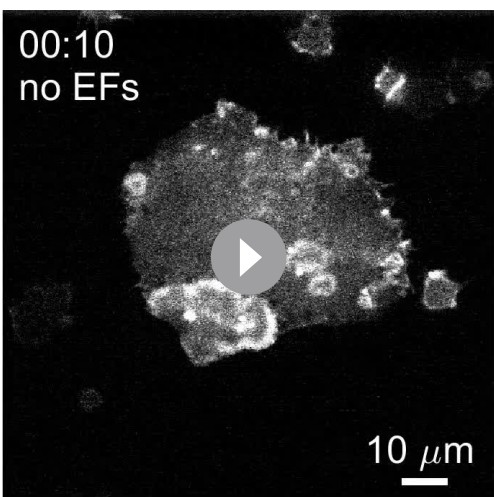

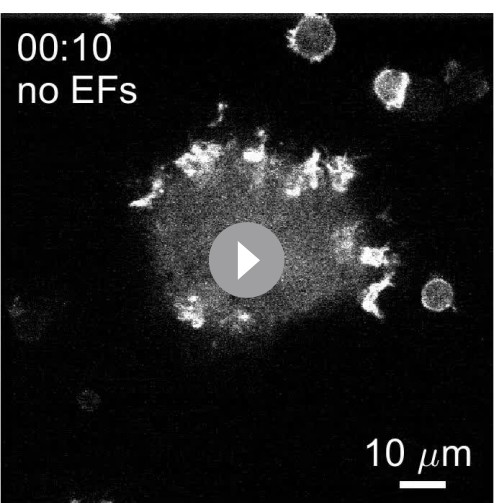

**Video 10.** Time-lapse confocal videos of limE-RFP on the basal membrane of a giant *Dictyostelium discoideum* cells set on a flat surface. An electric field of 20 V/cm was applied at 10 min, where the cathode was set at the left side. The cathode was reversed at 30 min. Images were acquired every 10 s and shown at five frames/s.

https://elifesciences.org/articles/73198/figures#video10

**Video 11.** Time-lapse confocal videos of limE-RFP on the dorsal membrane of a giant *Dictyostelium discoideum* cell set on a flat surface. An electric field of 20 V/cm was applied at 5 min, where the cathode was set at the left side. The cathode was reversed at 30 min. Images were acquired every 10 s and shown at 5 frames/s.

https://elifesciences.org/articles/73198/figures#video11

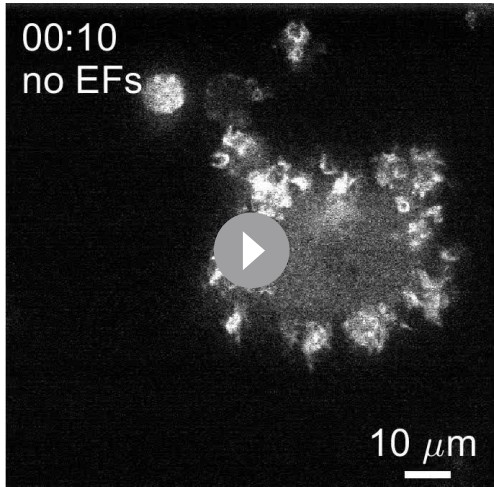

**Video 12.** Time-lapse confocal videos of limE-RFP on the dorsal membrane of a giant *Dictyostelium discoideum* cell set on a ridged surface. An electric field of 20 V/cm was applied at 10 min, where the cathode was set at the left side. The cathode was reversed at 30 min. Images were acquired every 10 s and shown at 5 frames/s.
https://elifesciences.org/articles/73198/figures#video12

direction on a time scale of ~7 min (*Figure 4e and f*). Our analysis further shows that larger waves in the old fronts are less sensitive to EF reversal than those in the new fronts (*Figure 4—figure supplement 2*).

## Discussion

By employing giant cells, in which the cortical waves are disentangled from cell motion, we demonstrate that EFs modulate cortical wave dynamics directly, providing a mechanism for cell guidance by EFs (*Figure 1c, b and d*). Our use of nanoridges to generate quasi-1D waves that are small, short-lived, and unable to turn (*Figure 1c*) enabled detailed quantification of wave properties, demonstrating that EFs directly affect the abundance, locations, and directions of cortical waves.

### EFs guide cortical wave dynamics

Previous studies have suggested that the basal cortical waves in *D. discoideum* are insensitive to external chemotactic gradients, whereas 'pseudopods' at other regions in the same cells can be guided (*Lange et al., 2016*). This conclusion is surprising because the biochemical events traveling with the waves are the same as those occurring on pseudopods, and pseudopods with the dorsal cups on the same cells do respond to chemoattractants. Also, similar cortical waves in human mammary epithelial cells can be guided effectively by epidermal growth factors (*Zhan et al., 2020*). Additional input from the greater contact of giant *D. discoideum* cells with the surface may outweigh the effect of applied chemical gradients on the basal waves. Other studies have shown that single cells can integrate combinations of external chemical and mechanical stimuli.

Our work shows that in giant cells, waves of both F-actin polymerization (*Figure 3*) and its upstream regulator PIP3 (*Figure 1—figure supplement 1*) are indeed guided by EFs. These biased biochemical and biomechanical events lead to more protrusions at the cell front than at the cell back, thus driving cell migration (*Figure 2—figure supplement 2*). The development of the biased wave activities takes ~10 min following the introduction of an EF (*Figures 2a and 3*), which is much slower than the timescale of surface-receptor-regulated chemotaxis. The high resistance of the cell membrane limits the effects of EFs on intracellular components, but EFs may act on the charged lipids and molecular clusters. Thus, we suspect that the slow response results from the electrophoresis of the charged membrane components involved in wave formation, which has a characteristic time scale of 5–10 min (*Allen et al., 2013*; *McLaughlin and Poo, 1981*).

We further explored the dynamics in response to EF reversal at the subcellular level using nanotopography (*Figure 4*). We observed that the new waves are induced to propagate towards the current cathode within 2–3 min (*Figure 4e* and *Figure 4—figure supplement 2*), suggesting that waves themselves can adapt quickly to the changing electrical environments. Because we only observed the fast adaptation on ridged surfaces, this phenomenon may be related to the shorter wave lifetimes on nanoridges than on flat surfaces. A short lifetime allows waves to be nucleated at a higher rate on the nanoridges, leading to a rapid directional response. During this process, the EF may regulate the wave nucleation through locally changing specific charged lipids, ion fluxes, or local pH gradients (*Crevenna et al., 2013*; *Frantz et al., 2008*; *Köhler et al., 2012*; *Martin et al., 2011*; *Zhou and Pang, 2018*).

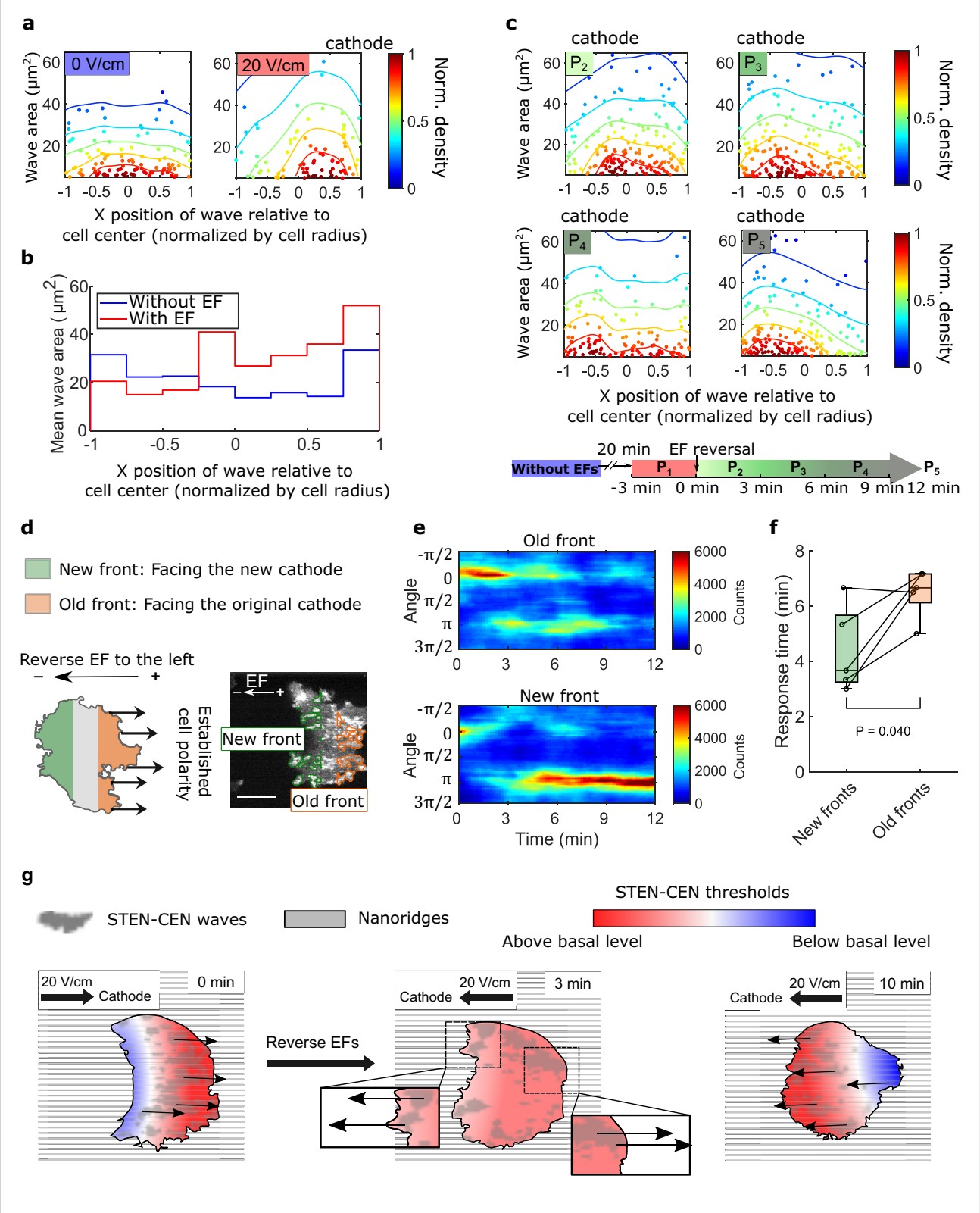

**Figure 4.** Spatial inhomogeneity of the response to EFs on nanoridges.
(**a**) Density scatter plots of the wave area vs. x position of the wave relative to the cell center. Nanoridges and EF are orientated in the x-direction. The difference of x coordinates of cell center and wave location was calculated, then the value was further normalized by the cell radius. Each point represents a wave, and all the points were collected from five independent experiments. The left plot is for a period in which there was no EF ($N_{wave}$

*Figure 4 continued on next page*

*Figure 4 continued*

= 296), and the right plot is for a period in which there was a 20 V/cm EF, during which the cells exhibited steady directional migration ($N_{wave}$ = 224). For each experiment without an EF, the EF was always turned on several minutes later. Thus, we defined the direction in which cathode was located in the presence of an EF as the positive direction in the absence of an EF. The color code corresponds to the density of points. (**b**) Average wave area in subcellular regions. The points in a were sectioned, based on their x position relative to the cell center (normalized by cell radius) at a bin size of 0.25 (8 sections in total from –1–1), and calculated the average wave area in each section. (**c**) Changes in actin waves' spatial distribution in response to EF reversal; data from six independent experiments. The color of each plot is coded according to the timeline displayed at the bottom of the panel. $P_2$-$P_5$: The EF was reversed, and cells gradually developed polarization toward the new cathode. The number of waves in each plot: $N_{p2}$ = 272, $N_{p3}$ = 277, $N_{p4}$ = 193, $N_{p5}$ = 246. (**d**) A schematic illustrating the old and new fronts of giant cells when the EF was reversed. (**e**) Time stacks of orientation distributions of optical-flow vectors at an old front and a new front. The EF was reversed from the cathode being at the right (0) to the cathode being at the left ($\pi$) at 0 min. (**f**) Comparisons of response time between new fronts (green) and old fronts (orange) from multiple experiments ($N_{cell}$ = 5). The p-value was calculated using a pairwise t-test at the 5% significance level. (**g**) Cartoon illustrating different time scales of local wave propagation and global rearrangement of STEN-CEN thresholds, in response to EF reversal.

The online version of this article includes the following source data and figure supplement(s) for figure 4:

**Source data 1.** Wave area and wave x position relative to the cell center (normalized by cell radius) in different periods of electrotaxis experiment.

**Figure supplement 1.** The spatial inhomogeneity of wave properties could be caused either by the EF or by the external electrical potential gradient relative to the ground.

**Figure supplement 1—source data 1.** Wave area and wave position relative to the center of microscopic field of view.

**Figure supplement 2.** Analysis of the propagation of individual waves.

**Figure supplement 2—source data 1.** Wave propagation direction in different periods of electrotaxis experiment Related to *Figure 4—figure supplement 2b*.

## EFs modulate the thresholds of the excitable wave system

Recent studies have shown that the cortical wave system can be described as a coupled signal transduction and cytoskeletal excitable network. Based on both simulation and experimental studies (*Bhattacharya et al., 2020*; *Miao et al., 2017*), it has been shown that the wave ranges, durations, and speeds are determined by the local threshold of activation, which in turn are regulated by the relative levels of activators and inhibitors (*Miao et al., 2017*; *Miao et al., 2019*).

Our quantification shows that guided waves become larger, faster, and more persistent in an EF (*Figure 2*), indicating that the excitable system is closer to its threshold for activation (*Miao et al., 2019*). This effect may arise from enhanced positive feedback, reduced negative feedback, or both. We further find that wave nucleation is enhanced at the cell front and suppressed at the back (*Figure 4a and b*). This subcellular inhomogeneity is consistent with a biased excitable network framework (*Iglesias and Devreotes, 2012*; *Meinhardt, 1999*; *Tang et al., 2014*; *Xiong et al., 2010*), which was added to the STEN-CEN model to introduce an internal spatial gradient in the local threshold of wave initiation, akin to cell polarity.

Local excitation and global inhibition (*Xiong et al., 2010*), LEGI, schemes have effectively recreated the features of both fast directional sensing and stable polarity in response to chemical signals, which can lead to robust biased excitable network. Both directional sensing and stable polarity can lead to a robust biased excitable network. For chemical signals, the directional response from PIP3 occurs within seconds, whereas the establishment of stable polarity usually requires many minutes. However, based on our analysis, establishing both directional response (*Figure 3*) and polarity (*Figure 4*) in response to EFs requires 5–10 min. It is worth noting that PIP3 waves also sense EFs on a time scale of minutes (*Figure 1—figure supplement 1*). Our observation suggests that EFs act on the polarity establishment rather than directional sensing. This hypothesis is supported by a recent study showing that G-protein-coupled receptors (GPCRs), which are the regulator in the LEGI model for *D. discoideum* that allows for sensing chemoattractant on timescales of seconds, are not essential for electrotaxis (*Zhao et al., 2002*).

## EFs act on waves, and waves determine cell behaviors

Our results raise the possibility that cortical wave dynamics are modulated directly by EFs and that the waves in turn mediate cellular response. Waves travel across cell membranes to coordinate the trailing edge with the front edge, and the cytoskeletal components in cortical waves are involved in developing the stable polarity. On the other hand, the duration and turning capacity of STEN-CEN

waves directly impact the speed and characteristics of the cellular response to EFs (*Figure 3*) on a longer timescale than that of surface-receptor-regulated chemotaxis.

Our results shed light on how EFs modulate protrusions. Previous studies have shown that various protrusions that drive cell motion, such as filopodia, lamellipodia (*Miao et al., 2019*), and macropinocytotic cups (*Video 4*), are always associated with expanding waves near cell perimeter. Our previous work has shown that changing wave properties by perturbing STEN-CEN states leads to the transition of protrusion profiles, which indicates that wave properties dictate the properties of the protrusions (*Miao et al., 2019*). Here we showed that EFs can alter the waves differently on the two ends of the cell (*Figure 4a*). As a result of these spatially inhomogeneous wave properties, protrusions become more abundant and larger on one side of the cell versus the other, which eventually leads to guidance of cell migration.

On flat surfaces, a slow U-turn is observed following EF reversal, whereas on nanoridges, faster switching is observed. Thus, the response of migrating cells to a changing guidance cue can be predicted from the characteristics of the waves driving the migration process. Indeed, the U-turn behaviors of neutrophils and differentiated, single *D. discoideum* cells in response to EF reversal (*Hind et al., 2016*; *Sato et al., 2007*; *Srinivasan et al., 2003*; *Xu et al., 2003*), which are usually ascribed to stable cell polarity, may instead reflect the persistence and 2D turning behavior of cortical waves in these environments (*Figure 3*).

Nanoridges allow us to shed further light on the multiscale character of the system, because cells include both short, 1D waves on the basal plane, and longer lasting, 2D waves on the dorsal plane. The different response times on the subcellular level due to different wave behaviors (*Figure 4—figure supplement 2*) provide strong evidence that cortical waves act as direct mediators of EFs. Waves on different planes are similar in composition but are impacted differently by the EF. We observed fast switching of wave directions in the basal plane near the ridged substrate and slower turning of the waves in the dorsal plane within the same cell, indicating that the direction of waves is controlled locally by external cues (*Figure 3—figure supplement 1*).

EFs provide a means to modulate cortical waves directly. On the other hand, biological conditions that modulate wave characteristics may also speed up or suppress the cellular response to directional cues. Longer-lasting waves offer persistence in the face of rapidly changing gradients, whereas shorter waves yield faster adaptability to changing directional signals. The durations of waves and their ability to turn together have a dominant effect on the response of cells to an EF.

## Materials and methods

**Key resources table**

| Reagent type (species) or resource | Designation | Source or reference | Identifiers | Additional information |
|---|---|---|---|---|
| Cell line (*D. discoideum*) | Aca null | https://doi.org/10.1016/S0092-8674(03)00081–3 | | The cell line was a gift from Carole A. Parent lab. |
| Cell line (*D. discoideum*) | PHcrac-GFP LimE-RFP | https://doi.org/10.1038/ncb3495 | | The cell line was a gift from Peter N. Devreotes lab. |
| Software, algorithm | Optical flow analysis (run by MATLAB) | https://doi.org/10.1091/mbc.E19-11-0614 | | |

### Cell line

In the study, we used LimE-RFP aca null *Dictyostelium* discoideum (D.d.) and PHcrac-GFP/LimE-RFP D.d cell lines. LimE-RFP aca null was a gift from Carole A. Parent lab (https://doi.org/10.1016/S0092-8674(03)00081-3), and PHcrac-GFP-LimE-RFP was a gift from Peter N. Devreotes lab (https://doi.org/10.1038/ncb3495). We have conducted the mycoplasma contamination testing for both cell lines and did not detected contamination.

### Cell culture

*Dictyostelium discoideum* cell lines were grown axenically in the HL5 medium. Aggregation adenylyl cyclase null (ACA−) mutants, which do not produce cAMP and do not have chemotaxis signal relay (*Kriebel et al., 2003*), were used in electrotaxis experiments to avoid chemotaxis. The cells used

also express limE-RFP as a reference for filamentous actin structures. G418 was used as the selection medium during cell culture. For the experiments in *Figure 1*, *Figure 1—figure supplement 1*, we used *Dictyostelium discoideum* co-expressing PHcrac-GFP and LimE-RFP, and we used G418 as the selective medium. Note that an enhancement in LimE concentration is associated with protrusions, as seen in *Figure 2—figure supplement 2*, and that the protrusions are biased to the side facing the cathode. Because protrusions are driven by F-actin polymerization, we believe this observation rules out the possibility that LimE binding/unbinding to/from F-actin itself is sensitive to EFs.

## Electrofusion

Cells were washed twice with 17 mM Sorensen buffer (15 mM $KH_2PO_4$ and 2 mM $Na_2HPO_4$, pH 6.0) and rolled for 30 min at a concentration of $1.5 \times 10^7$ mL$^{-1}$. Electrofusion was conducted with a Gene Pulser Gen1 system. Three pulses of 1 kV at a 1 s interval were applied. After electroporation, cells were relaxed for 5 min. Then cells were diluted to $5 \times 10^5$ mL$^{-1}$ with normal developing buffer (5 mM $KH_2PO_4$, 5 mM $Na_2HPO_4$, 2 mM $MgCl_2$ and 0.2 mM $CaCl_2$, PH 6.5) and seeded into a customized electrotactic chamber, with dimensions 20 mm × 5 mm × 0.25 mm. The aca null cell line that we used did not generate many waves in the vegetative stage, and electro-fusion with 1 kV pulses stressed the cells. Thus cells were starved for 2 hr before experiments to generate more actin waves.

## Nanotopography fabrication

The nanotopographic pattern used in these cell studies was fabricated through a technique known as multiphoton absorption polymerization (MAP), as described elsewhere (*LaFratta et al., 2006*; *LaFratta et al., 2004*). An ultrafast, pulsed laser beam (Coherent Mira 900 F, 76 MHz) was passed through a high-numerical-aperture microscope objective into a photopolymerizable resin sandwiched between glass coverslips. A LabVIEW (National Instruments) program allowed for control of the stage position and the shutter state, determining where polymerization occurred (and did not) in the resin, allowing patterning. Once fabrication was completed, the patterned sample was developed in ethanol twice for 3 min each to remove unreacted monomer. The polymerized structure was baked at 110 °C for at least 1 hr.

To produce the necessary number of replicate patterns with the same dimensions, an adapted version of replica molding was performed (*Sun et al., 2018*). A hard polydimethylsiloxane (*h*-PDMS) film containing hexanes to increase the resolution of feature replication was spin-coated onto the functionalized structure made from MAP. The film was allowed to sit on the structure for 2 hr at room temperature and was then baked at 60 °C for 1 hr. Regular PDMS (Sylgard 184) was prepared at a 10:1 ratio of elastomer base to the curing agent by degassing and mixing. The PDMS was poured onto the *h*-PDMS film, and molding was completed by baking at 60 °C for an additional 70 min. The final mold was peeled from the glass slide supporting the MAP-patterned structure.

The mold was used to produce replicas of the original pattern. A drop of the same acrylic resin was placed on the patterned area of the PDMS mold, and then an acrylate-functionalized glass coverslip was pressed firmly on top, spreading the sandwiched drop. Tape secured this system in place. The resin was cured for a total of 5 min under a UV lamp (Blak-ray), producing a polymer film. It should be noted that the PDMS mold is the negative relief pattern of the structure made using MAP. Therefore, samples (or replicas) of the original pattern could be produced on a relatively large scale with this method. The replicas were soaked in ethanol for at least 12 hr before use in the cell studies. We fabricated samples with flat surfaces by using a PDMS mold with a smooth surface.

## Lattice light-sheet microscopy

The 3i lattice light-sheet microscope in the Johns Hopkins School of Medicine Microscope Facility was used for two-color, 3D imaging. Vegetative, single *Dictyostelium* cells were seeded on a circular 5 mm coverslip patterned with nanoridges, which was immersed in a bath of standard developing buffer throughout imaging.

## Electrotaxis experiments

We 3D-printed electrotaxis chambers (*Figure 2—figure supplement 1*) with dimensions of 20 mm × 5 mm × 0.25 mm and composed of a clear resin using a Formlabs Form2 3D-printer. Agar bridges were used to isolate cell media from electrodes to minimize electrochemical products and pH changes.

Twenty V/cm constant EFs were applied. Time-lapse images of the phase-contrast channel and the RFP/GFP channel were recorded using PerkinElmer spinning-disk microscope at a frame rate of 0.1 frames/s (Yokogawa CSU-X1 spinning-disk scan head (5000 rpm)) with Hamamatsu EMCCD camera and Volocity analysis software.

## Optical-flow analysis and model fitting of actin polymerization dynamics

We applied the Lukas-Kanade optical-flow method to quantify the direction of the intensity flow in fluorescence videos. This algorithm produced pixel-basis vector fields of intensity motion. Before applying the optical-flow algorithm, each image was smoothed by a 2D Gaussian filter ($\sigma$ = 3) to reduce noise. After the smoothing, we further removed the flow vectors created by noise using optical-flow reliability as our criterion. The reliability is defined as the smallest eigenvalues of the $A^TwA$ matrix, where $w$ is a Gaussian weight matrix and $A$ is the intensity gradient matrix. The size of the weight matrix for *D. discoideum* was set at 19 × 19, with standard deviation $\sigma$ = 2 (0.42 μm).

We built a bimodal von Mises model to compare the actin and cellular responses accurately. A von Mises distribution is given by

$$f_{VM}\left(\theta|\mu,\kappa\right) = \frac{e^{\kappa\cos(\theta-\mu)}}{2\pi I_0(\kappa)} \quad . \tag{2}$$

where the peak location is $\mu$ and the concentration $\kappa$. The orientation distribution of optical-flow vectors at each time point is fit with two von Mises distributions

$$f\left(\theta|\mu_1,\mu_2,\kappa,p_1,p_2\right) = p_1 f_{VM}\left(\theta|\mu_1,\kappa\right) + p_2 f_{VM}\left(\theta|\mu_1 + \pi,\kappa\right) \tag{3}$$

where $p$ is the proportion of each component. We use the constraints

$$\mu_1 - \mu_2 = \pi \tag{4}$$

and

$$p_1 + p_2 = 1 \tag{5}$$

Maximum likelihood estimation (MLE) is applied to estimate model parameters based on the orientation of all the optical-flow vectors every 12 frames (2 min). With this model, we can quantitatively study the temporal change of actin dynamics. The preferential direction is defined as the $\mu$ with the largest proportion $p$ at each time point.

## Quantification of actin wave properties

The segmentation of actin waves was conducted based on the combined information from fluorescence intensities and optical flow. We first applied the kmeans ($k$ = 3) cluster (*Kanungo et al., 2002*) to pick up the bright regions in the limE-RFP videos, then only kept the moving objects by applying the reliability mask from the optical-flow analysis.

To classify the large actin structures composed of substructures moving independently, we considered the optical flow at the edges of the large structure (*Figure 2c*). If the optical-flow vectors were moving in the same direction at both edges, the large structure was classified as a single wave. Otherwise, the large structure was divided into multiple smaller patches. In the latter case, a pronounced boundary was detectable between two substructures, then we used the detected boundary to divide the large structure into multiple substructures.

After classification, we measured properties such as wave speed, wave duration, and wave area to characterize STEN-CEN. Wave speed was measured by tracking the clusters of optical-flow vectors oriented in similar directions. The detailed algorithm can be found in a prior publication (*Lee et al., 2020*). To measure wave duration, we first tracked actin waves using a customized, multi-object tracking tool based on the overlapping areas between frames. A unique identification number was assigned to each wave, then wave duration, wave area (measured by the Matlab function regionprops) were recorded for each wave.

## Acknowledgements

We thank Q Qing and M Zhao for discussions.

## Additional information

### Funding

| Funder | Grant reference number | Author |
|---|---|---|
| Air Force Office of Scientific Research | FA9550-16-1-0052 | Qixin Yang Yuchuan Miao Leonard J Campanello Matt J Hourwitz Bedri Abubaker-Sharif Abby L Bull Peter Devreotes John T Fourkas Wolfgang Losert |
| National Institute of Health | T32 GM136577 | Bedri Abubaker-Sharif |

The funders had no role in study design, data collection and interpretation, or the decision to submit the work for publication.

### Author contributions

Qixin Yang, Data curation, Formal analysis, Investigation, Visualization, Writing - original draft, Writing – review and editing; Yuchuan Miao, Investigation, Visualization, Writing – review and editing; Leonard J Campanello, Software; Matt J Hourwitz, Investigation, Writing – review and editing; Bedri Abubaker-Sharif, Investigation, Methodology; Abby L Bull, Writing – review and editing; Peter N Devreotes, John T Fourkas, Wolfgang Losert, Conceptualization, Supervision, Writing – review and editing

### Author ORCIDs

Qixin Yang http://orcid.org/0000-0001-5349-6992
Bedri Abubaker-Sharif http://orcid.org/0000-0001-5003-7934
John T Fourkas http://orcid.org/0000-0002-4522-9584
Wolfgang Losert http://orcid.org/0000-0002-1792-7860

### Decision letter and Author response

Decision letter https://doi.org/10.7554/eLife.73198.sa1
Author response https://doi.org/10.7554/eLife.73198.sa2

## Additional files

### Supplementary files

• Transparent reporting form

### Data availability

The data and the codes that were used to analyze the data have been uploaded to Dyrad. https://doi.org/10.5061/dryad.f7m0cfxx4.

The following dataset was generated:

| Author(s) | Year | Dataset title | Dataset URL | Database and Identifier |
|---|---|---|---|---|
| Yang Q, Miao Y, Hourwitz M, Campanello L, Bull A, Devreotes P, Fourkas J, Losert W | 2021 | Cortical waves mediate the cellular response to electric fields | https://dx.doi.org/10.5061/dryad.f7m0cfxx4 | Dryad Digital Repository, 10.5061/dryad.f7m0cfxx4 |

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
