## [Editor Report]

The authors combine a series of clever biological approaches to fuse small *Dictyostelium* cells into "giant cells" that greatly facilitate the spatial resolution of actin wave dynamics without or with electrical stimulation when grown on smooth or nano-textured surfaces. This compelling experimental system opens possibilities for the field to analyze the molecular subtleties involved in these cytoskeletal reorganizations.

---

## [Decision Letter]

**Decision letter after peer review:**

Thank you for submitting your article "Cortical waves mediate the cellular response to electric fields" for consideration by *eLife*. Your article has been reviewed by 3 peer reviewers, one of whom is a member of our Board of Reviewing Editors, and the evaluation has been overseen by Jonathan Cooper as the Senior Editor. The reviewers have opted to remain anonymous.

Essential revisions:

*eLife* usually recommends to include a numbered list of essential revision requirements. However, in the case of your manuscript, the 3 reviewers, who appreciated your work, all raised different and interesting points that deserve a response from you. Therefore, we are exceptionally leaving the comments of the different reviewers for you to answer separately. Since many of these points do not require any experimental effort from you, this should allow you to improve your manuscript with a reasonable amount of work for these revisions.*Reviewer #1 (Recommendations for the authors):*

I would recommend its publication provided that few complementary experiments are performed to validate this approach.

Main comments:

1/ How good is LimE-RFP as a marker of actin networks? Is there any evidence from the literature that LimE-RFP binding/unbinding to/from actin filaments is not itself sensitive to EF? If this was the case, it would complexify greatly the analysis of the data. If no information is available from the literature, the authors should perform control experiments with a different actin marker (ideally GFP-actin expressed at low level) to confirm this point unambiguously.

2/ Response to EF could have been characterized even better. What happens when EF are switched off? Is the process of network growth reversible and at which time scale?

In Figure 2a, could the authors provide comparative data in the absence of EF?

Is the effect of network propagation under EF saturating after a certain time (> 25 min)?

What is the threshold intensity of EF that is required to initiate visible wave propagation in this system?

*Reviewer #2 (Recommendations for the authors):*

Several aspects should be addressed and clarified.

1) The authors convincingly show that the dynamics of cortical waves can be guided by EFs. But it remains unclear to me what actually is the role of cortical waves in electrotaxis. Does electrotaxis depend on cortical waves or do we also observe electrotaxis in cells that don't show any waves? D. discoideum cells do not show waves at all times. Also, there are cell lines that do not show basal actin waves. So this could be actually tested, unless the authors want to concentrate on the wave dynamics and leave their functional role in electrotaxis open for now. The wave dynamics are very interesting in its own right but a clearer statement regarding the role of cortical waves in electrotaxis would give their findings a wider biological relevance.

2) A very nice aspect of the present work is that not only basal but also dorsal waves are considered. However, some of the results on dorsal waves do not look convincing to me. In Figure S2, it is shown that conclusions about the dynamics of dorsal waves are drawn from 2D images taken along a planar confocal cut (red line in Figure S2). This means that features on top of the cell (above this plane) may be missed, and looking at the corresponding videos, it indeed is obvious that most of the activity is seen along the "border", where the cut intersects with the dorsal membrane. Why are the authors not using the full power of their lattice light-sheet microscope and image the dorsal side in full 3D? Is this technically not possible? It would be much clearer and more convincing to actually track the dorsal waves across the full dorsal membrane and not only at the rim, similar to waves at the basal membrane.

3) In the Discussion, the authors interpret their findings in the context of excitable waves (STEN-CEN waves). If this way of modeling cortical waves is correct, I understand that the present findings can be probably related to this concept. However, the line of argments is somewhat vague and hand-waving because the authors do not present an actual extension of their STEN-CEN model to incorporate electric fields. For example, I am not sure if waves necessarily will become "larger, faster, and more persistent" if the excitable system is closer to its activation threshold. Perhaps the authors could demonstrate this based on their existing STEN-CEN framework?

Line 340: it is not clear to me why “the coexistence of waves with different behaviors in different portions of the cell provides powerful evidence that cortical waves act as direct mediators of Efs.” What is meant by “direct mediators”? Earlier (line 242) you say that it is the polarized intracellular environment that causes the spatial inhomogeneity in the response to Efs. So aren’t the waves rather “indirect mediators” of the Efs? Please explain.

References:

There is quite a bit of literature on actin waves in D. discoideum. In addition to Ref. 8 which addresses oversized cells, I suggest that you give more credit to earlier work by Gunther Gerisch and others, also with respect to possible functions of actin waves in motility, division etc.

*Reviewer #3 (Recommendations for the authors):*

The big picture summary is found in the public review. Here, we list the specific comments we have that are needed to strengthen the work for eventual publication. I have kept experimental requests to a minimum, and indicated the points where they are discussed with an (X) symbol.

1. (X) Lack of experimental methods clarity. In a field as confusing as electrotaxis/galvanotaxis, it is not acceptable to simply write in the methods: “20V/cm constant EF fields were applied.” How were they applied? Which specific approach was deployed and what did the chamber look like, and how might the choice of system have affected the results? 20V/cm is a very high stimulus for injecting charge into an aqueous electrolyte medium—surely this would produce hydrolysis and pH changes unless this is controlled for? All of these concerns can be addressed by a proper discussion of the methodology. However, this reviewer feels two experimental details (X) are required here. (1) Why was 20 V/cm chosen, and how does the cortical wave phenotype vary with field strength? What happens if a lower or higher field strength is used? It seems like these results would greatly help bolster the STEN-CEN framework and clarify the input-output aspects. (2) Can the authors validate that they controlled well for pH changes due to electrolysis, or might this be an underlying mechanism?

2. Statistics. Many figure panels are missing error bars, and there are very few statistical tests or direct comparisons to help understand how different given phenotypes actually are or how representative the presented data are. Figures 2b,f ; 3c,d; and 4b seem like obvious cases where more discussion of the stats is needed. Similarly, it was often unclear how many samples were being compared in those same panels. A clearer discussion in the text, captions, methods, and those figure panels emphasizing how many explicit experimental replicates and explicit individual cells analyzed would go a long way towards clarifying that these data are truly representative. Control cases were also lacking in a number of figures (see Figure discussion below).

3. The STEN-CEN framework was exciting (pun intended), but also didn’t seem to coalesce into a clear finding. It’s a great idea to try to link these findings to an excitable system with known dynamics, but STEN, CEN, BEN and other acronyms were frequently deployed without obvious quantitative connections to the data itself, at least in the sense that non-specialist readers such as myself would be able to follow this. Given the centrality of STEN-CEN to the introduction and claims in the discussion, I think the authors need to do better represent the importance and validity of STEN-CEN here. I understand that Figure 1 shows ‘STEN-CEN’ dynamics, but most of that discussion took the form of (“see Figure 1b 150s-200s”) and it wasn’t really clear to me what I should be looking for in these panels. If Figure 1 is the place where STEN-CEN is validated as being relevant, it would be great to see something a little clearer in the text, and in the figure emphasizing the specific characteristics we should focus on.

4. The nano-ridges seemed almost superfluous until the end. The nano-ridges are quite clever, but their introduction and discussion of their importance seemed a bit scattered, despite their being involved in each figure. I would have appreciated a clearer discussion of why this method was necessary compared to just the smooth surface data. Esotaxis is interesting, but it wasn’t clear that it was necessary for most of the claims here, although I can see how it was useful. Kudos for the detailed methods section here. One experimental question—did the use of PDMS here affect dicty motion relative to migration on glass/plastic in the flat surface cases?

5. (X) Biological claims and mechanisms. This is a nice biophysics paper-I do not think the authors need to go further down the rabbit hole and prove a specific molecular mechanism. However, I do think they need to be a little more careful about some of the claims and to also spend more time discussing *why* they think these waves are forming here and what might be nucleating the waves upstream of them. Speculation would OK here, and I felt the paper sort of overemphasized the dynamics of the waves without really musing on how the sausage was made. More discussion would help here. That said, there is one biological experiment that seems easily within reach and, unless I misread the paper, is missing. The authors stated up front that the co-localization of F-actin and PIP3 in *control* data meant that only F-actin needed to be evaluated for the actual stimulation experiments. It was actually unclear from Figure 1 how strong that co-localization actually was (see below), but more to the point-I think the authors need (X) to show even a single supplemental figure clearly demonstrating that this overlap of PIP3/F-actin still holds in the stimulated case as well as the control rather than simply assuming that it does from looking only at the control data. Apologies if I misinterpreted the data and this is already accounted for; it just wasn’t clear to me and I’d like to see an analogous panel to some of the data currently in Figure 1 except for stimulated cells.

Figure 1. Why is red consistently ‘outlining’ green, and how does that support the co-localization claim?

Figure 2d/e were a little hard to assess in the sense that going from 0-20V/cm seems like it should produce a drastic change, but the visualizations seem relatively similar, so I think I’m not reading these right. Can these be clarified in the figure or caption? 2f doesn’t have any statistics (I get that it’s a histogram, but I don’t know how important the implied shift in the peaks is).

Figure S4-I did not follow this discussion. How do the authors use the data in S4 to show that polarized intracellular environments were key vs. external fields? I was also confused about that claim because I thought it was well established that the field must act on the extracellular membrane components because it can’t penetrate the plasma membrane (re: impedance), so I wasn’t sure what this discussion point or figure were addressing.

---

## [Author Response]

Reviewer #1 (Recommendations for the authors):I would recommend its publication provided that few complementary experiments are performed to validate this approach.Main comments:1/ How good is LimE-RFP as a marker of actin networks? Is there any evidence from the literature that LimE-RFP binding/unbinding to/from actin filaments is not itself sensitive to EF? If this was the case, it would complexify greatly the analysis of the data. If no information is available from the literature, the authors should perform control experiments with a different actin marker (ideally GFP-actin expressed at low level) to confirm this point unambiguously.

Thank you for the recommendation to publish the manuscript and for this important question of whether we are observing a bias in actin polymerization or merely a bias in actin odelling. We did not find any literature that addresses whether LimE binding / unbinding is sensitive to Efs. However, we see that an enhancement in LimE concentration is associated with protrusions, as seen in supplementary Figure 2 —figure supplement 2. The protrusions are biased to the side facing the cathode. Because protrusions are driven by F-actin polymerization, we would expect no bias in protrusions if our observations were based on a odelling bias. Also, as explained in our response to the next comment, the EF-induced changes in actin polymerization persist for some time after the field is turned off, suggesting that the field itself is not directly altering the binding of LimE to F-actin. We have added the explanation to the cell culture method session.

Edited methods section of cell culture

“Note that an enhancement in LimE concentration is associated with protrusions, as seen in Figure 2 —figure supplement 2, and that the protrusions are biased to the side facing the cathode. Because protrusions are driven by F-actin polymerization, we believe this observation rules out the possibility that LimE binding/unbinding to/from F-actin itself is sensitive to Efs.”

2/ Response to EF could have been characterized even better. What happens when EF are switched off? Is the process of network growth reversible and at which time scale?

Thank you for these suggestions. We have conducted more experiments in which we turned off the EF after 30 min and recorded for an additional 20 min. We found that the actin-wave propagation direction became unbiased within 4 to 6 min. However, we also found that the higher concentration of polymerized actin observed in the presence of an EF remained elevated throughout the 20 min time window after the EF was switched off (not shown in the published paper but shown to the reviewer). Such long-term effects of an EF point to a complex adaptation of cells to the EF, which is beyond the scope of the current manuscript.

In Figure 2a, could the authors provide comparative data in the absence of EF?

We updated figures 2a and b with the results of new control experiments in which we recorded ode for 30 min in the absence of an EF. In the absence of an EF, we did not observe a significant increase in polymerized actin.

Is the effect of network propagation under EF saturating after a certain time (> 25 min)?

In Figure 3d, we show that the wave dynamics become stable after ~ 15 min of turning on the EF.

What is the threshold intensity of EF that is required to initiate visible wave propagation in this system?

We have carried out additional experiments with lower Efs and found that giant cells did not respond to a 10 V/cm EF but did respond to a 15 V/cm EF. Thus, we estimated the threshold intensity of EF is in between 10 V/cm and 15 V/cm. We describe this result in the revised manuscript and include as new supplementary material videos of giant cells in the presence of a 10 V/cm EF (Video7) and a 15 V/cm EF (Video 8).

Added to Results section:

“We found that giant cells respond to a narrow range (15 V/cm to 20 V/cm) of EF amplitudes (Video 7 and Video 8), and that higher voltage (35 V/cm) damaged cells.”

Reviewer #2 (Recommendations for the authors):Several aspects should be addressed and clarified.1) The authors convincingly show that the dynamics of cortical waves can be guided by Efs. But it remains unclear to me what actually is the role of cortical waves in electrotaxis. Does electrotaxis depend on cortical waves or do we also observe electrotaxis in cells that don’t show any waves? D. discoideum cells do not show waves at all times. Also, there are cell lines that do not show basal actin waves. So this could be actually tested, unless the authors want to concentrate on the wave dynamics and leave their functional role in electrotaxis open for now. The wave dynamics are very interesting in its own right but a clearer statement regarding the role of cortical waves in electrotaxis would give their findings a wider biological relevance.

We have reworded our discussion (see below) to explain better the role of cortical waves during electrotaxis. We agree that there are cells in which large ventral waves are not visible, but these cells likely have “normal” sized waves associated with each protrusion. Devreotes’ previous studies have shown that the protrusion activities are always associated with wave activities. If a cell has no waves (such as in cells treated with signal transduction inhibitors), it cannot move in a dc EF. A study on cells without large ventral waves is an exciting topic, but it is beyond the scope of this paper, and we would like to leave it for future studies.

Edited Discussion section

“Efs act on waves, and waves determine cell behaviors

Our results raise the possibility that cortical wave dynamics are modulated directly by Efs and that the waves in turn mediate cellular response. Waves travel across cell membranes to coordinate the trailing edge with the front edge, and the cytoskeletal components in cortical waves are involved in developing the stable polarity. On the other hand, the duration and turning capacity of STEN-CEN waves directly impact the speed and characteristics of the cellular response to Efs (Figure 3) on a longer timescale than that of surface-receptor-regulated chemotaxis.

Our results shed light on how Efs modulate protrusions. Previous studies have shown that various protrusions that drive cell motion, such as filopodia, lamellipodia (Miao et al., 2019), and macropinocytotic cups (Video 4), are always associated with expanding waves near cell perimeter. Our previous work has shown that changing wave properties by perturbing STEN-CEN states leads to the transition of protrusion profiles, which indicates that wave properties dictate the properties of the protrusions (Miao et al., 2019). Here we showed that Efs can alter the waves differently on the two ends of the cell (Figure 4a). As a result of these spatially inhomogeneous wave properties, protrusions become more abundant and larger on one side of the cell versus the other, which eventually leads to guidance of cell migration.

On flat surfaces, a slow U-turn is observed following EF reversal, whereas on nanoridges, faster switching is observed. Thus, the response of migrating cells to a changing guidance cue can be predicted from the characteristics of the waves driving the migration process. Indeed, the U-turn behaviors of neutrophils and differentiated, single D. discoideum cells in response to EF reversal (Hind et al., 2016; Sato et al., 2007; Srinivasan et al., 2003; Xu et al., 2003), which are usually ascribed to stable cell polarity, may instead reflect the persistence and 2D turning behavior of cortical waves in these environments (Figure 3).

Nanoridges allow us to shed further light on the multiscale character of the system, because cells include both short, 1D waves on the basal plane, and longer-lasting, 2D waves on the dorsal plane. The different response times on the subcellular level due to different wave behaviors (Figure 4 —figure supplement 2) provide strong evidence that cortical waves act as direct mediators of Efs. Waves on different planes are similar in composition but are impacted differently by the EF. We observed fast switching of wave directions in the basal plane near the ridged substrate and slower turning of the waves in the dorsal plane within the same cell, indicating that the direction of waves is controlled locally by external cues (Figure 3 —figure supplement 1).

Efs provide a means to modulate cortical waves directly. On the other hand, biological conditions that modulate wave characteristics may also speed up or suppress the cellular response to directional cues. Longer-lasting waves offer persistence in the face of rapidly changing gradients, whereas shorter waves yield faster adaptability to changing directional signals. The durations of waves and their ability to turn together have a dominant effect on the response of cells to an EF.”

2) A very nice aspect of the present work is that not only basal but also dorsal waves are considered. However, some of the results on dorsal waves do not look convincing to me. In Figure S2, it is shown that conclusions about the dynamics of dorsal waves are drawn from 2D images taken along a planar confocal cut (red line in Figure S2). This means that features on top of the cell (above this plane) may be missed, and looking at the corresponding videos, it indeed is obvious that most of the activity is seen along the “border”, where the cut intersects with the dorsal membrane. Why are the authors not using the full power of their lattice light-sheet microscope and image the dorsal side in full 3D? Is this technically not possible? It would be much clearer and more convincing to actually track the dorsal waves across the full dorsal membrane and not only at the rim, similar to waves at the basal membrane.

Thank you for the suggestion. We agree that our data is just a cross-section of the actual waves, and full z-scanning using lattice light-sheet would be more powerful than our spinning disk experiments. Unfortunately, the lattice light-sheet system sample chamber is not compatible with the channel device used to generate reproducible Efs. Photo-bleaching and laser damage prevented capturing more than two z planes and tracking waves along the curved surface for our spinning-disk system. However, these cross-sections did provide some information, such as where waves are located, and in which direction the waves move in the x-y plane. Accordingly, we have added explanation to the text and legends indicating that the ventral and dorsal data are not exactly comparable:

Edited text (current Figure 3 – supplement figure 1)

“Figure 3 —figure supplement 1. Basal actin waves reverse direction on nanoridges, whereas dorsal waves turn. A. A schematic showing the two imaging planes used, with the morphology of the substrate. B. LimE-RFP images recorded at the dorsal plane. Unlike the basal focal plane images, which capture the complete basal wave dynamics, the dorsal plane images do not capture the full dorsal wave motion. To avoid photobleaching and laser damage, we only imaged the cross-section of the dorsal waves and tracked the cross-sections using optical-flow analysis”

3) In the Discussion, the authors interpret their findings in the context of excitable waves (STEN-CEN waves). If this way of odelling cortical waves is correct, I understand that the present findings can be probably related to this concept. However, the line of argments is somewhat vague and hand-waving because the authors do not present an actual extension of their STEN-CEN model to incorporate electric fields.

We have worked on the discussion to do a better job of relating the STEN-CEN model with our findings. We did not extend our STEN-CEN model with Efs in the current manuscript. Still, there are numerous works on the extended STEN-CEN model with LEGI/BEN to explain how the excitable system is modulated by external chemical cues, the mechanism of which is similar to electrotaxis to some extent. We built up our discussion of how Efs perturb STEN-CEN based on the current chemotaxis models.

Discussion section

“Efs modulate the thresholds of the excitable wave system

Recent studies have shown that the cortical wave system can be described as a coupled signal transduction and cytoskeletal excitable network. Based on both simulation and experimental studies (Bhattacharya et al., 2020; Miao et al., 2017), it has been shown that the wave ranges, durations, and speeds are determined by the local threshold of activation, which in turn are regulated by the relative levels of activators and inhibitors (Miao et al., 2017, 2019).

Our quantification shows that guided waves become larger, faster, and more persistent in an EF (Figure 2), indicating that the excitable system is closer to its threshold for activation (Miao et al., 2019). This effect may arise from enhanced positive feedback, reduced negative feedback, or both. We further find that wave nucleation is enhanced at the cell front and suppressed at the back (Figure 4a, b). This subcellular inhomogeneity is consistent with a biased excitable network framework (Iglesias and Devreotes, 2012; Meinhardt, 1999; Tang et al., 2014; Xiong et al., 2010), which was added to the STEN-CEN model to introduce an internal spatial gradient in the local threshold of wave initiation, akin to cell polarity.

Local excitation and global inhibition (Xiong et al., 2010), LEGI, schemes have effectively recreated the features of both fast directional sensing and stable polarity in response to chemical signals, which can lead to robust biased excitable network. Both directional sensing and stable polarity can lead to a robust biased excitable network. For chemical signals, the directional response from PIP3 occurs within seconds, whereas the establishment of stable polarity usually requires many minutes. However, based on our analysis, establishing both directional response (Figure 3) and polarity (Figure 4) in response to Efs requires 5 to 10 min. It is worth noting that PIP3 waves also sense Efs on a time scale of minutes (Figure 1 —figure supplement 1). Our observation suggests that Efs act on the polarity establishment rather than directional sensing. This hypothesis is supported by a recent study showing that G-protein-coupled receptors (GPCRs), which are the regulator in the LEGI model for D. Discoideum that allows for sensing chemoattractant on timescales of seconds, are not essential for electrotaxis (Zhao et al., 2002).”

For example, I am not sure if waves necessarily will become “larger, faster, and more persistent” if the excitable system is closer to its activation threshold. Perhaps the authors could demonstrate this based on their existing STEN-CEN framework?

We now clarify that previous studies combining simulations and experiments have shown that the thresholds of the excitable system determine wave properties. Please refer to the last section discussion for more details.

Line 340: it is not clear to me why "the coexistence of waves with different behaviors in different portions of the cell provides powerful evidence that cortical waves act as direct mediators of EFs." What is meant by "direct mediators"?

We have rephrased the sentence to make it clearer.

Edited text

“The different response times on the subcellular level due to different wave behaviors (Figure 4 —figure supplement 2) provide strong evidence that cortical waves act as direct mediators of EFs. Waves on different planes are similar in composition but are impacted differently by the EF. We observed fast switching of wave directions in the basal plane near the ridged substrate and slower turning of the waves in the dorsal plane within the same cell, indicating that the direction of waves is controlled locally by external cues (Figure 3 —figure supplement 1).”

Earlier (line 242) you say that it is the polarized intracellular environment that causes the spatial inhomogeneity in the response to EFs. So aren't the waves rather "indirect mediators" of the EFs? Please explain.

Based on our data, we showed that there were both fast, local responses and slow, global responses to EFs (Figure 4). We showed that the polarized intracellular environment caused slow global responses (Figure 4). In contrast, our analysis on single-wave dynamics showed that waves themselves displayed faster response than cellular level response (Figure 4 —figure supplement 2). These two results suggest that waves themselves sense EFs rapidly but that polarity also interacts with wave dynamics and leads to slower global rearrangement. We have included the Discussion section in response to major comment 5 from reviewer 3. Please see the third paragraph in that Discussion section for the actual edits.

References:There is quite a bit of literature on actin waves in D. discoideum. In addition to Ref. 8 which addresses oversized cells, I suggest that you give more credit to earlier work by Gunther Gerisch and others, also with respect to possible functions of actin waves in motility, division etc.

Thank you for the suggestion. We have edited our reference and added more references about actin waves driving migration and division.

Edited text

“Actin polymerization, coordinated with its associated signaling molecules, self-organizes into microscale spatial regions that travel as waves across plasma membranes. These waves drive various cell behaviors, such as migration and division (Bhattacharya et al., 2019; Bretschneider et al., 2009; Flemming et al., 2020; Gerhardt et al., 2014; Gerisch, 2010).”

Reviewer #3 (Recommendations for the authors):The big picture summary is found in the public review. Here, we list the specific comments we have that are needed to strengthen the work for eventual publication. I have kept experimental requests to a minimum, and indicated the points where they are discussed with an (X) symbol.1. (X) Lack of experimental methods clarity. In a field as confusing as electrotaxis/galvanotaxis, it is not acceptable to simply write in the methods: "20V/cm constant EF fields were applied." How were they applied? Which specific approach was deployed and what did the chamber look like, and how might the choice of system have affected the results?

We made a new supplemental figure (Figure 2 —figure supplement 1) to illustrate our experiment setup. This device is optimized for easy assembly and minimal leakage based on a previously published protocol paper (Yang et al., 2014).

20V/cm is a very high stimulus for injecting charge into an aqueous electrolyte medium--surely this would produce hydrolysis and pH changes unless this is controlled for? All of these concerns can be addressed by a proper discussion of the methodology.

We isolated the cell media from the electrodes using agar bridges to avoid electrochemical products and pH changes. Please check the new figure supplement Figure 2 —figure supplement 1 for the setup.

Edited method section

Electrotaxis experiments

“We 3D-printed electrotaxis chambers (Figure 2 —figure supplement 1) with dimensions of 20 mm × 5 mm × 0.25 mm and composed of a clear resin using a Formlabs Form2 3D-printer. Agar bridges were used to isolate cell media from electrodes to minimize electrochemical products and pH changes. 20 V/cm constant EFs were applied. Time-lapse images of the phase-contrast channel and the RFP/GFP channel were recorded using PerkinElmer spinning-disk microscope at a frame rate of 0.1 frames/s (Yokogawa CSU-X1 spinning-disk scan head (5000 rpm)) with Hamamatsu EMCCD camera and Volocity analysis software.”

However, this reviewer feels two experimental details (X) are required here. (1) Why was 20 V/cm chosen, and how does the cortical wave phenotype vary with field strength? What happens if a lower or higher field strength is used? It seems like these results would greatly help bolster the STEN-CEN framework and clarify the input-output aspects.

We chose 20 V/cm because we found the dynamical range of electrotaxis was narrow: we have tried lower voltages and found giant cells respond to EFs in the range of 15 V/cm to 20 V/cm. Higher field strengths (e.g., 35 V/cm) damaged cells.

Edited result section

“We found that giant cells respond to a narrow range (15 V/cm to 20 V/cm) of EF amplitudes (Video 7 and Video 8), and that higher voltage (35 V/cm) damaged cells.”

(2) Can the authors validate that they controlled well for pH changes due to electrolysis, or might this be an underlying mechanism?

We isolated the cell media from the electrodes using agar bridges to avoid electrochemical products and pH changes. Please check the new figure supplement Figure 2 – supplement figure 1 for the setup.

2. Statistics. Many figure panels are missing error bars, and there are very few statistical tests or direct comparisons to help understand how different given phenotypes actually are or how representative the presented data are. Figures 2b,f ; 3c,d; and 4b seem like obvious cases where more discussion of the stats is needed. Similarly, it was often unclear how many samples were being compared in those same panels. A clearer discussion in the text, captions, methods, and those figure panels emphasizing how many explicit experimental replicates and explicit individual cells analyzed would go a long way towards clarifying that these data are truly representative. Control cases were also lacking in a number of figures (see Figure discussion below).

Thank you for bringing up this concern. We have added more precise statistical analysis to the panels you mentioned and clarified the number of samples/independent experiments, the statistical methods in the figure captions, and method sessions (Please see the specific locations in the following ‘figure specific comments’). We also added the control cases in the figures.

3. The STEN-CEN framework was exciting (pun intended), but also didn't seem to coalesce into a clear finding. It's a great idea to try to link these findings to an excitable system with known dynamics, but STEN, CEN, BEN and other acronyms were frequently deployed without obvious quantitative connections to the data itself, at least in the sense that non-specialist readers such as myself would be able to follow this. Given the centrality of STEN-CEN to the introduction and claims in the discussion, I think the authors need to do better represent the importance and validity of STEN-CEN here.

Thank you for the suggestion. We now explain the relevance of the STEN-CEN model in the introduction by explaining prior studies of the Devreotes group that used the model to explain cell protrusions and cell migration. We then introduce BEN/LEGI in the discussion, where it becomes necessary to make sense of our results.

We also revised the discussion to relate our results in Figure 2 and Figure 4 more closely to the STEN-CEN model. Please find the Discussion section in the previous major comment 3 from reviewer 2.

Edited introduction section related to STEN-CEN

“Actin polymerization, coordinated with its associated signaling molecules, self-organizes into microscale spatial regions that travel as waves across plasma membranes. These waves drive various cell behaviors, such as migration and division (Bhattacharya et al., 2019; Bretschneider et al., 2009; Flemming et al., 2020; Gerhardt et al., 2014; Gerisch, 2010). The wave system can be described as a coupled signal transduction excitable network – cytoskeletal excitable network (STEN-CEN) (Devreotes et al., 2017; Miao et al., 2019). STEN-CEN has the characteristics of an excitable system, including exhibiting an activation threshold for wave initiation and experiencing refractory periods. It has been shown that the STEN-CEN wave properties dictate protrusion properties (Miao et al., 2019). Tuning the activity levels of key components in STEN-CEN changes wave patterns, which leads to the transition of protrusion profiles. An activator/inhibitor, reaction/diffusion system model successfully recapitulates the experimental results (Bhattacharya et al., 2020; Bhattacharya and Iglesias, 2018). For simplicity, here we will refer to STEN-CEN waves as cortical waves.”

I understand that Figure 1 shows 'STEN-CEN' dynamics, but most of that discussion took the form of ("see Figure 1b 150s-200s") and it wasn't really clear to me what I should be looking for in these panels. If Figure 1 is the place where STEN-CEN is validated as being relevant, it would be great to see something a little clearer in the text, and in the figure emphasizing the specific characteristics we should focus on.

We edited the text and captions related to figure 1 to highlight the features of interest of the excitable system, that is, wave duration, wave sizes, and the existence of refractory periods. We also worked on the related discussion paragraphs.

Edited Result Section related to the key features of STEN-CEN

“In giant cells, multiple waves were initiated randomly and propagated radially across the basal membranes (Figure 1b and Video 2). CEN is driven by STEN, but has a substantially shorter characteristic timescale. Thus, PIP3 waves displayed band-like shapes, whereas F-actin appeared across the bands with higher levels at the rims of PIP3 waves (Miao et al., 2019). As shown in Figure 1b, colliding waves did not cross, but instead rotated by 90° (Figure 1b, 150 s – 200 s). This behavior is suggestive of a refractory period following excitation, which is a hallmark of an excitable system. On nanoridges, the giant cells generated multiple, quasi-1D patches of F-actin and PIP3 with shorter lifetimes than on flat surfaces (Figure 1c and Video 3). Some waves formed and propagated for a short distance (Line 2 in Figure 1c), whereas others formed and then quickly dissipated (Line 3 in Figure 1c). The wave dissipation can be explained in terms of an excitable system with lateral inhibition, in which the dispersion of the inhibitor is faster than that of the activator. Thus, the waves eventually dissipate due to the spatial accumulation of the inhibitor. Prior studies have shown that in this situation, the excitable system threshold determines the wave duration (Bhattacharya et al., 2020; Ermentrout et al., 1984). As was the case on flat surfaces, 1D patches occurred throughout the basal surfaces on ridges, and thus were independent of cell motion.”

4. The nano-ridges seemed almost superfluous until the end. The nano-ridges are quite clever, but their introduction and discussion of their importance seemed a bit scattered, despite their being involved in each figure. I would have appreciated a clearer discussion of why this method was necessary compared to just the smooth surface data. Esotaxis is interesting, but it wasn't clear that it was necessary for most of the claims here, although I can see how it was useful.

We have revised the introduction to clarify that nanotopography is a tool for us based on our extensive prior studies but is not the main focus of the current work (Edited text 1).

Edited introduction section related to nanotopography

“We further use nanotopography to alter the waves’ spatial structures and characteristic timescales. Upon contact with nanotopography, cells produce quasi-1D wave patches. The phenomenon of guided actin polymerization by nanotopography is known as esotaxis (Driscoll et al., 2014), which has been investigated in detail (Ketchum et al., 2018; Lee et al., 2020). There are several advantages of incorporating nanotopography in our study. First, these waves persist for a shorter time on nanotopography than on flat surfaces, enabling us to investigate whether wave systems with different characteristic timescales respond to EFs differently. Second, waves on ridged surfaces have shorter lifetimes than those on flat surfaces, and thus only propagate in local regions of giant cells. Therefore, nanotopography allows us to distinguish between local and global mediation of the EF response.”

Kudos for the detailed methods section here. One experimental question--did the use of PDMS here affect dicty motion relative to migration on glass/plastic in the flat surface cases?

We now clarify that PDMS is the mold for the final replicas we used. We added a paragraph describing how we fabricate nanotopography from the PDMS mold. The material for nanotopography is an acrylic resin, and all the flat surfaces we used in this study are also made of the same resin.

Methods section

“The mold was used to produce replicas of the original pattern. A drop of the same acrylic resin was placed on the patterned area of the PDMS mold, and then an acrylate-functionalized glass coverslip was pressed firmly on top, spreading the sandwiched drop. Tape secured this system in place. The resin was cured for a total of 5 min under a UV lamp (Blak-ray), producing a polymer film. It should be noted that the PDMS mold is the negative relief pattern of the structure made using MAP. Therefore, samples (or replicas) of the original pattern could be produced on a relatively large scale with this method. The replicas were soaked in ethanol for at least 12 h before use in the cell studies. We fabricated flat surface samples by using a PDMS mold with a smooth surface.”

5. (X) Biological claims and mechanisms. This is a nice biophysics paper-I do not think the authors need to go further down the rabbit hole and prove a specific molecular mechanism. However, I do think they need to be a little more careful about some of the claims and to also spend more time discussing why they think these waves are forming here and what might be nucleating the waves upstream of them. Speculation would OK here, and I felt the paper sort of overemphasized the dynamics of the waves without really musing on how the sausage was made. More discussion would help here. That said, there is one biological experiment that seems easily within reach and, unless I misread the paper, is missing. The authors stated up front that the co-localization of F-actin and PIP3 in control data meant that only F-actin needed to be evaluated for the actual stimulation experiments. It was actually unclear from Figure 1 how strong that co-localization actually was (see below), but more to the point-I think the authors need (X) to show even a single supplemental figure clearly demonstrating that this overlap of PIP3/F-actin still holds in the stimulated case as well as the control rather than simply assuming that it does from looking only at the control data. Apologies if I misinterpreted the data and this is already accounted for; it just wasn't clear to me and I'd like to see an analogous panel to some of the data currently in Figure 1 except for stimulated cells.

Thank you for the suggestion. We have carried out new experiments to image both LimE and PHcrac in the presence of EFs, and significantly revised the Discussion sections. We included a supplementary figure (Figure 1 – supplement figure 1) to show PIP3 and F-actin are still coupled in the presence of EFs.

Result section related to imaging both PIP3 and F-actin

“PIP3 activity was coordinated with F-actin activity (Profiles in Figure 1a, 1b, and 1c), both in the absence (Figure 1) and presence of an EF (Figure 1 —figure supplement 1, Video 6). Therefore, in the experiments described below, we only monitored F-actin activity.”

Edited Discussion section

“EFs guide cortical wave dynamics

Previous studies have suggested that the basal cortical waves in D. discoideum are insensitive to external chemotactic gradients, whereas “pseudopods” at other regions in the same cells can be guided (Lange et al., 2016). This conclusion is surprising because the biochemical events traveling with the waves are the same as those occurring on pseudopods, and pseudopods with the dorsal cups on the same cells do respond to chemoattractants. Also, similar cortical waves in human mammary epithelial cells can be guided effectively by epidermal growth factors (Zhan et al., 2020). Additional input from the greater contact of giant D. discoideum cells with the surface may outweigh the effect of applied chemical gradients on the basal waves. Other studies have shown that single cells can integrate combinations of external chemical and mechanical stimuli.

Our work shows that in giant cells, waves of both F-actin polymerization (Figure 3) and its upstream regulator PIP3 (Figure 1 —figure supplement 1) are indeed guided by EFs. These biased biochemical and biomechanical events lead to more protrusions at the cell front than at the cell back, thus driving cell migration (Figure 2 —figure supplement 2). The development of the biased wave activities takes ~10 min following the introduction of an EF (Figure 2a and Figure 3), which is much slower than the timescale of surface-receptor-regulated chemotaxis. The high resistance of the cell membrane limits the effects of EFs on intracellular components, but EFs may act on the charged lipids and molecular clusters. Thus, we suspect that the slow response results from the electrophoresis of the charged membrane components involved in wave formation, which has a characteristic time scale of 5 to 10 min (Allen et al., 2013; McLaughlin and Poo, 1981).

We further explored the dynamics in response to EF reversal at the subcellular level using nanotopography (Figure 4). We observed that the new waves are induced to propagate towards the current cathode within 2 to 3 min (Figure 4e and Figure 4 —figure supplement 2), suggesting that waves themselves can adapt quickly to the changing electrical environments. Because we only observed the fast adaptation on ridged surfaces, this phenomenon may be related to the shorter wave lifetimes on nanoridges than on flat surfaces. A short lifetime allows waves to be nucleated at a higher rate on the nanoridges, leading to a rapid directional response. During this process, the EF may regulate the wave nucleation through locally changing specific charged lipids, ion fluxes, or local pH gradients (Crevenna et al., 2013; Frantz et al., 2008; Köhler et al., 2012; Martin et al., 2011; Zhou and Pang, 2018).”

Figure specific comments:Figure 1. Why is red consistently 'outlining' green, and how does that support the co-localization claim?

This phenomenon of red outlining green is a characteristic observation in prior studies of this system and can be understood through the STEN-CEN model. We have added more explanation about STEN-CEN in the Results section related to figure 1.

Result section related to Figure 1

“In giant cells, multiple waves were initiated randomly and propagated radially across the basal membranes (Figure 1b and Video 2). CEN is driven by STEN, but has a substantially shorter characteristic timescale. Thus, PIP3 waves displayed band-like shapes, whereas F-actin appeared across the bands with higher levels at the rims of PIP3 waves (Miao et al., 2019).”

Figure 2d/e were a little hard to assess in the sense that going from 0-20V/cm seems like it should produce a drastic change, but the visualizations seem relatively similar, so I think I'm not reading these right. Can these be clarified in the figure or caption?

20 V/cm did produce a drastic change, as shown in Figure 2a, in the sense that larger waves are generated. However, because one large wave involves much more actin polymerization than one small wave, the number of large waves is limited compared to the number of small waves. In Figure 2d/e, each dot represents a wave, and it is expected that the pattern is determined by the many small waves (red region). If you compare the areas representing large wave sizes (yellow-blue regions), there are indeed more waves in a dc EF. To make it clearer, we have also included the average of areas and dimensions in the text related to Figure 2d/e.

Edited text related to Figure 2d/e

“Density scatter plots of both dimensions exhibit elliptical contours (Figure 2d), suggesting that nanotopography constrains wave growth. With an EF parallel to the ridges, the waves broadened in both directions (Figure 2d). The average increases in wave dimension parallel and perpendicular to the ridges were 20% and 13%, respectively, and the average increase in wave area was 44%.”

Figure S4-I did not follow this discussion. How do the authors use the data in S4 to show that polarized intracellular environments were key vs. external fields? I was also confused about that claim because I thought it was well established that the field must act on the extracellular membrane components because it can't penetrate the plasma membrane (re: impedance), so I wasn't sure what this discussion point or figure were addressing.

Because the giant cells span ~50 mm, we wanted to ensure the intracellular bias was caused by the EF, not the absolute potential relative to the ground. The result was expected. Hence we only included the analysis as supplementary material.

Edited result section related to Figure S4 (current version Figure 4 – supplement figure 1)

“We also measured the wave properties in the single cells scattered throughout the field of view but did not observe a corresponding gradient of wave properties among single cells closer to the cathode versus the cells closer to the anode. This result indicates that the spatial inhomogeneity shown in Figures 4a, b was caused by the EF rather than by the absolute electrical potential relative to the ground (Figure 4 —figure supplement 1).”

Edited Figure 4 – supplement figure 1 caption

“The spatial inhomogeneity of wave properties could be caused either by the EF or by the external electrical potential gradient relative to the ground. To explore these scenarios, we quantified the waves in single cells surrounding the giant cell in the field of view, where the center of the field was defined as the origin. In contrast to the giant cells, these single cells are scattered throughout the field of view but are not large enough for the potential gradient to create significant intracellular polarization. Thus, if the spatial inhomogeneity is caused by the external electrical potential gradient, we would observe a gradient of wave properties from single cells located in the region between -50 µm and 50 µm. a. A limE image. Single cells are highlighted with blue circles. b, c. Density scatter plots of wave location vs. wave area for single cells (highlighted by blue circles in a) in the absence (b) and the presence (c) of EF. Unlike in giant cells (Figure 4a, 4b), the wave areas in single cells are spatially homogeneous. The ratio of mean wave areas in the regions nearer the cathode (location > 0) to those in regions farther from the cathodes (location < 0) was calculated for both single cells and giant cells. In giant cells, this ratio increases from 0.85 to 2.00 with an EF (Figure 4a), whereas for single cells, the ratio is almost unchanged (1.08 without an EF and 0.98 with an EF). This analysis was based on the experiments from 4 different days.”

Figure 4. Lots going on in this figure! 4b-how many cells, what are the statistics, how representative is this? 4f-it seems like many cells must have been analyzed, so I'm curious why only 5 are shown here and it's not clear if the p value means this is a strong conclusion or not given only 5 datapoints.

We have edited the caption of figure 4 for clarity and to include information on the number of cells. We have several hundreds of waves in the 4a-4c wave-basis analysis, but we only have 5 giant cells for 4e-4f cell-basis analysis.

Edited Figure 4 caption

“Figure 4. Spatial inhomogeneity of the response to EFs on nanoridges. a. Density scatter plots of the wave area vs. x position of the wave relative to the cell center. Nanoridges and EF are orientated in the x-direction. The difference of x coordinates of cell center and wave location was calculated, then the value was further normalized by the cell radius. Each point represents a wave, and all the points were collected from 5 independent experiments. The left plot is for a period in which there was no EF (N_wave_ = 296), and the right plot is for a period in which there was a 20 V/cm EF, during which the cells exhibited steady directional migration (N_wave_ = 224). For each experiment without an EF, the EF was always turned on several minutes later. Thus, we defined the direction in which cathode was located in the presence of an EF as the positive direction in the absence of an EF. The color code corresponds to the density of points. b**.** Average wave area in sub-cellular regions. The points in a were sectioned, based on their x position relative to the cell center (normalized by cell radius) at a bin size of 0.25 (8 sections in total from -1 to 1), and calculated the average wave area in each section. c. Changes in actin waves' spatial distribution in response to EF reversal; data from 6 independent experiments. The color of each plot is coded according to the timeline displayed at the bottom of the panel. P_2_-P_5_: The EF was reversed, and cells gradually developed polarization towards the new cathode. The number of waves in each plot: N_p2_ = 272, N_p3_ = 277, N_p4_ = 193, N_p5_ = 246 d. A schematic illustrating the old and new fronts of giant cells when the EF was reversed. e. Time stacks of orientation distributions of optical-flow vectors at an old front and a new front. The EF was reversed from the cathode being at the right (0) to the cathode being at the left (p) at 0 min. f. Comparisons of response time between new fronts (green) and old fronts (orange) from multiple experiments (N_cell_ = 5). The P-value was calculated using a pairwise t-test at the 5% significance level. g. Cartoon illustrating different time scales of local wave propagation and global rearrangement of STEN-CEN thresholds, in response to EF reversal.”

Reference mentioned in the response

Yang, H. Y., La, T. D., and Isseroff, R. R. (2014). Utilizing custom-designed galvanotaxis chambers to study directional migration of prostate cells. *Journal of Visualized Experiments*, *94*, 1–8. https://doi.org/10.3791/51973